# SIRT1 regulates sphingolipid metabolism and neural differentiation of mouse embryonic stem cells through c-Myc-SMPDL3B

Wei Fan[1], Shuang Tang[1†], Xiaojuan Fan[2], Yi Fang[1], Xiaojiang Xu[3], Leping Li[4], Jian Xu[5], Jian-Liang Li[3], Zefeng Wang[2]*, Xiaoling Li[1]*

[1]Signal Transduction Laboratory, National Institute of Environmental Health Sciences, Research Triangle Park, Triangle Park, United States; [2]CAS Key Laboratory of Computational Biology, CAS-MPG Partner Institute for Computational Biology, Shanghai Institute of Nutrition and Health, Shanghai Institutes for Biological Sciences, University of Chinese Academy of Sciences, CAS Center for Excellence in Molecular Cell Science, Chinese Academy of Sciences, Shanghai, China; [3]Integrative Bioinformatics Support Group, National Institute of Environmental Health Sciences, Research Triangle Park, Triangle Park, United States; [4]Biostatistics & Computational Biology Branch, National Institute of Environmental Health Sciences, Research Triangle Park, Triangle Park, United States; [5]Children's Medical Center Research Institute, Department of Pediatrics, Harold C. Simmons Comprehensive Cancer Center, and Hamon Center for Regenerative Science and Medicine, University of Texas Southwestern Medical Center, Dallas, United States

*For correspondence:
wangzefeng@picb.ac.cn (ZW);
lix3@niehs.nih.gov (XL)

Present address: †Cancer Institute and Department of Nuclear Medicine, Fudan University Shanghai Cancer Center, Shanghai, China

Competing interests: The authors declare that no competing interests exist.

**Abstract** Sphingolipids are important structural components of cell membranes and prominent signaling molecules controlling cell growth, differentiation, and apoptosis. Sphingolipids are particularly abundant in the brain, and defects in sphingolipid degradation are associated with several human neurodegenerative diseases. However, molecular mechanisms governing sphingolipid metabolism remain unclear. Here, we report that sphingolipid degradation is under transcriptional control of SIRT1, a highly conserved mammalian $NAD^+$-dependent protein deacetylase, in mouse embryonic stem cells (mESCs). Deletion of SIRT1 results in accumulation of sphingomyelin in mESCs, primarily due to reduction of SMPDL3B, a GPI-anchored plasma membrane bound sphingomyelin phosphodiesterase. Mechanistically, SIRT1 regulates transcription of *Smpdl3b* through c-Myc. Functionally, SIRT1 deficiency-induced accumulation of sphingomyelin increases membrane fluidity and impairs neural differentiation in vitro and in vivo. Our findings discover a key regulatory mechanism for sphingolipid homeostasis and neural differentiation, further imply that pharmacological manipulation of SIRT1-mediated sphingomyelin degradation might be beneficial for treatment of human neurological diseases.

## Introduction

First isolated from brain extract in the 1880s, sphingolipid is a class of natural lipids containing a backbone of sphingoid base sphingosine (*Chen et al., 2010*; *Merrill et al., 2007*; *Pralhada Rao et al., 2013*). Sphingolipids not only are the structural components of cell membranes, but also act as signaling molecules to control various cellular events, including signal transduction, cell growth, differentiation, and apoptosis (*Hannun and Obeid, 2008*; *van Meer et al., 2008*). Particularly,

**eLife digest** All cells in the brain start life as stem cells which are yet to have a defined role in the body. A wide range of molecules and chemical signals guide stem cells towards a neuronal fate, including a group of molecules called sphingolipids. These molecules sit in the membrane surrounding the cell and play a pivotal role in a number of processes which help keep the neuronal cell healthy.

Various enzymes work together to break down sphingolipids and remove them from the membrane. Defects in these enzymes can result in excess levels of sphingolipids, which can lead to neurodegenerative diseases, such as Alzheimer's, Parkinson's and Huntington's disease. But how these enzymes are used and controlled during neuronal development is still somewhat of a mystery. To help answer this question, Fan et al. studied an enzyme called SIRT1 which has been shown to alleviate symptoms in animal models of neurodegenerative diseases.

Stem cells were extracted from a mouse embryo lacking the gene for SIRT1 and cultured in the laboratory. These faulty cells were found to have superfluous amounts of sphingolipids, which made their membranes more fluid and reduced their ability to develop into neuronal cells. Further investigation revealed that SIRT1 regulates the degradation of sphingolipids by promoting the production of another enzyme called SMPDL3B. Fan et al. also found that when female mice were fed a high-fat diet, this caused sphingolipids to accumulate in their embryos which lacked the gene for SIRT1; this, in turn, impaired the neural development of their offspring.

These findings suggest that targeting SIRT1 may offer new strategies for treating neurological diseases. The discovery that embryos deficient in SIRT1 are sensitive to high-fat diets implies that activating this enzyme might attenuate some of the neonatal complications associated with maternal obesity.

sphingolipids are enriched in microdomains/lipid rafts, a liquid-ordered phase in plasma membrane, where sphingomyelin, glycosphingolipids, and cholesterol form unique platforms for many different proteins that are important for nutrient transport, organelle contact, membrane trafficking, and homotypic fusion (*Brown and London, 1998*).

The homeostasis of sphingolipids is maintained by a highly coordinated metabolic network that links together various pathways with ceramide as a central node (*Figure 1—figure supplement 1A*). Firstly, ceramide can be de novo synthesized in endoplasmic reticulum (ER), starting from the condensation of serine and fatty acyl-CoA by serine palmitoyltransferase (SPT). Ceramide is then transported to the Golgi complex and further converted into more complex forms of sphingolipids, such as glycosphingolipids or sphingomyelin. Secondly, ceramide can be regenerated by hydrolysis of complex sphingolipids in the Golgi complex and plasma membrane, which is catalyzed by a class of specific hydrolases and phosphodiesterases. For example, regeneration of ceramide from sphingomyelin can be mediated by plasma-membrane-bound sphingomyelin phosphodiesterases (SMPDs), including SMPDL3B, a GPI-anchored lipid raft SMPD (*Heinz et al., 2015*). By degrading sphingomyelin, SMPDL3B regulates plasma membrane fluidity, which in turn impacts TLR-mediated innate immunity in macrophages (*Heinz et al., 2015*) and modulates insulin receptor signaling in podocytes (*Mitrofanova et al., 2019*). Sphingolipid metabolism is highly sensitive to environmental/nutritional perturbations. Sphingolipid biosynthesis and accumulation can be induced by high-fat diet (HFD) feeding in multiple tissues in mice (*Choi and Snider, 2015*). Sphingolipids are also accumulated during aging (*Giusto et al., 1992*; *Lightle et al., 2000*), and caloric restriction, a dietary regimen known to extend life span and delay a number of age-associated diseases, decreases sphingolipid accumulation by reducing the activity of SPT (*Tacconi et al., 1991*). Dietary serine restriction also alters sphingolipid diversity and synthesis to constrain tumor growth (*Muthusamy et al., 2020*).

Disruption of sphingolipid homeostasis has been involved in the pathogenesis of a number of human diseases, such as Niemann-Pick disease and neurodegenerative Alzheimer's, Parkinson's, and Huntington's diseases. These human diseases are generally the outcomes of defect in enzymes that degrade the sphingolipids (*Brice and Cowart, 2011*; *Alvarez-Vasquez et al., 2005*). Such degradation defect leads to accumulation of sphingolipids, which in turn dramatically alters cellular membrane structure and signaling, thereby triggering various diseases. Consequently, manipulation of

one enzyme or metabolite in sphingolipid degradation pathways may result in unexpected changes in cellular metabolic programs and related cellular functions (*Brice and Cowart, 2011*; *Alvarez-Vasquez et al., 2005*). However, molecular mechanisms that regulate the expression of these sphingolipid degrading enzymes remain unclear.

In the present study, we investigated the role of SIRT1 in regulation of sphingolipid degradation in mouse embryonic stem cells (mESCs) and mouse embryos. SIRT1 is an $NAD^+$-dependent protein deacetylase critical for multiple cellular processes, including metabolism, inflammation, stress response, and stem cell functions (*Houtkooper et al., 2012*; *Tang et al., 2014*; *Han et al., 2008*). SIRT1 is also a key regulator of animal development (*McBurney et al., 2003*; *Cheng et al., 2003*; *Wang et al., 2008*) and is particularly important in the central nervous system. For instance, SIRT1 has a major influence on hypothalamic function, and cell-type specific SIRT1 mutations result in defects in systemic energy metabolism, circadian rhythm, and the lifespan of the animal (*Dietrich et al., 2010*; *Ramadori et al., 2010*; *Ramadori et al., 2011*; *Satoh et al., 2013*). SIRT1 also modulates dendritic and axonal growth (*Hisahara et al., 2008*; *Michán et al., 2010*), and regulates synaptic plasticity and memory formation in adult brain (*Gao et al., 2010*). Moreover, SIRT1 has been shown to ameliorate neurodegenerative phenotypes in animal models of Alzheimer's, Parkinson's, and Huntington's disease (reviewed in *Herskovits and Guarente, 2014*). However, despite multiple mechanisms proposed for these critical roles of SIRT1 in the brain, how SIRT1 regulates neural development and functions remains unclear. Through global metabolomics and cellular metabolic characterizations, we discovered that SIRT1 deficiency in mESCs results in abnormal accumulation of sphingolipids, primarily due to reduced degradation of these lipids. We further found that this abnormal accumulation of sphingolipids does not affect the maintenance of pluripotent mESCs, but delays their neural differentiation during in vitro neural differentiation and in vivo mouse embryogenesis. Moreover, we provide evidence that SIRT1 regulates sphingolipid metabolism through deacetylation of c-Myc transcription factor, which promotes the expression of SMPDL3B and subsequent sphingomyelin degradation in mESCs. Together, our study identifies the SIRT1-c-Myc axis as an important regulatory mechanism for cellular sphingolipid metabolism and neural differentiation.

## Results

### Deletion of SIRT1 in ESCs results in accumulation of sphingomyelin

During a large-scale unbiased metabolomic analysis of WT and SIRT1 KO mESCs cultured in a serum-containing M10 medium, we discovered that SIRT1 KO mESCs display altered lipid metabolism, particularly metabolites involved in sphingolipid metabolism (*Figure 1A* and *Supplementary file 1*), in addition to previously reported metabolic defects in methionine metabolism (*Tang et al., 2017*). Specifically, SIRT1 KO mESCs had a dramatic accumulation of sphingomyelin in both complete medium and a methionine restricted medium (*Figure 1B*, *Figure 1—figure supplement 1B*, and *Supplementary file 1*). Moreover, deletion of SIRT1 in mel1 human ESCs by CRISPR/Cas9-mediated gene editing technology (*Figure 1—figure supplement 1C*) also led to accumulation of several types of sphingomyelin regardless of medium methionine contents (*Supplementary file 2*), indicating that SIRT1 regulates sphingolipid metabolism in ESCs independently of cellular methionine metabolism.

To confirm that deletion of SIRT1 indeed increases sphingomyelin contents in ESCs, we loaded WT and SIRT1 KO mESCs cultured in serum-free ESGRO medium with a green-fluorescent dye labeled sphingomyelin, BODIPY FL-$C_5$-sphingomyelin, for 30 min at 4°C, then chased at 37°C for 30 min. Both WT and SIRT1 KO mESCs were labeled with this green-fluorescent sphingomyelin (*Figure 1C*). However, SIRT1 KO mESCs had marked accumulation of BODIPY FL-$C_5$-sphingomyelin inside cells, presumably in ER and Golgi, compared to WT mESCs. Quantitative FACS analysis showed that SIRT1 KO mESCs have about a 50% increase in cellular levels of BODIPY FL-$C_5$-sphingomyelin compared to WT mESCs (*Figure 1D*). An enzyme-coupled colorimetric assay further revealed a ~60% increase of endogenous sphingomyelin in SIRT1 KO mESCs (*Figure 1E*). Interestingly, the accumulation of sphingomyelin was specific to mESCs, as SIRT1 KO MEFs had a comparable staining intensity of BODIPY FL-$C_5$-sphingomyelin as WT MEFs (*Figure 1—figure supplement 1D*). Additionally, SIRT1 KO mESCs had a similar staining intensity of BODIPY FL-$C_5$-Ceramide compared to WT

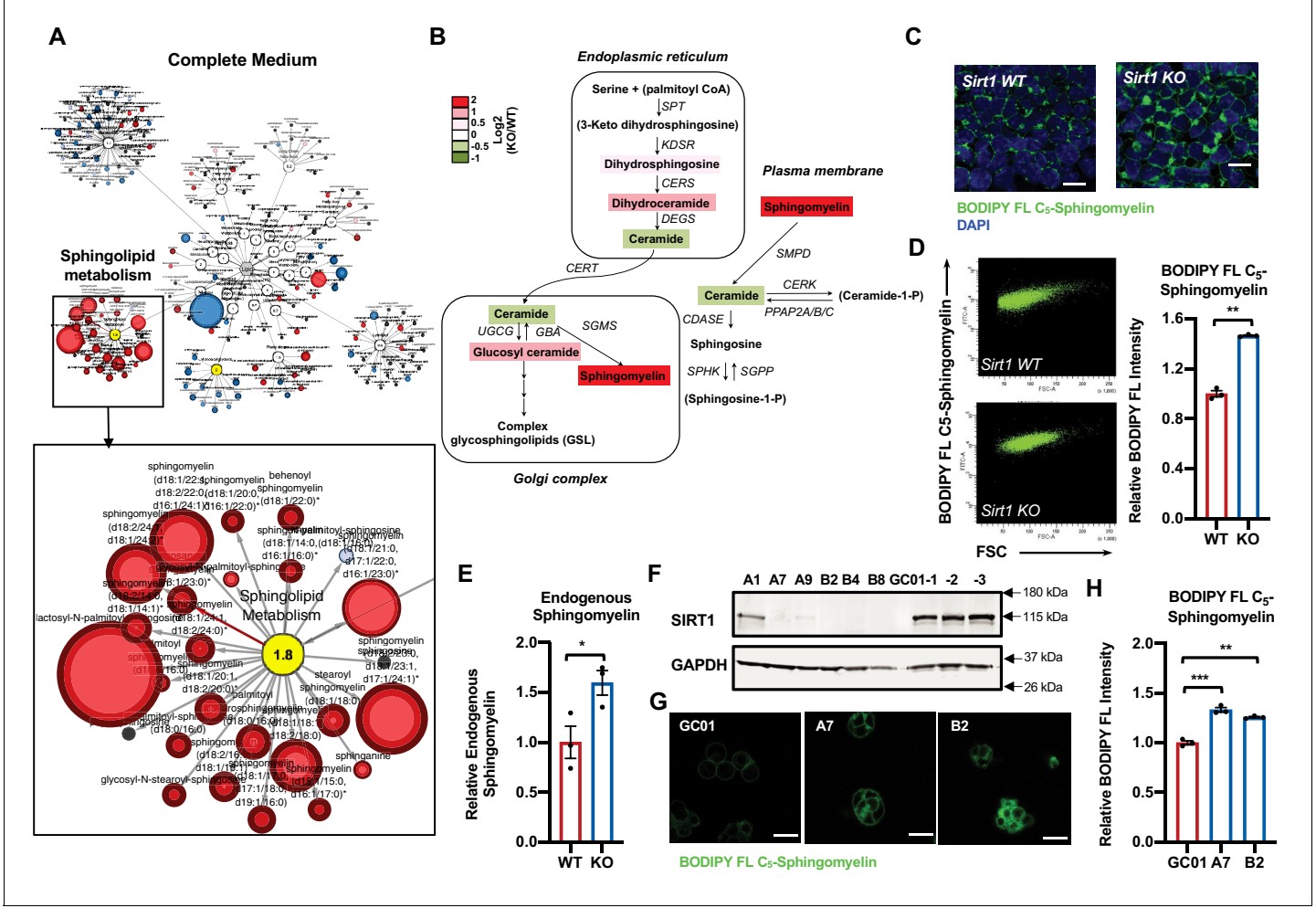

**Figure 1.** Deletion of SIRT1 in mESCs results in a dramatic accumulation of sphingomyelin. (A) Metabolomic analysis reveals a massive accumulation of sphingolipids in SIRT1 KO mESCs. WT and SIRT1 KO mESCs were cultured in a complete mESC maintenance medium (M10) and metabolites were analyzed by metabolomics as described in Materials and methods. The networks of significantly changed metabolites in lipid metabolism were analyzed by Cytoscape 2.8.3. Metabolites increased in SIRT1 KO mESCs were labeled red (p<0.05) or pink (0.05 < p < 0.10), metabolites decreased in SIRT1 KO mESCs were labeled blue (p<0.05) or light blue (0.05 < p < 0.10). Metabolite node size is proportional to the fold change in KO vs WT (n = 5 biological replicates). (B) The relative abundance of different metabolites mapped into sphingolipid metabolism pathways. Metabolites in sphingolipid metabolism in WT and SIRT1 KO mESCs were analyzed as in (A) and the relative abundance of metabolites involved in sphingolipid metabolism was displayed by the heat map (n = 5 biological replicates). (C–D) SIRT1 KO mESCs have increased levels of BODIPY FL-labeled sphingomyelin. WT and SIRT1 KO mESCs cultured in ESGRO medium were labeled with BODIPY FL-labeled sphingomyelin for 30 min at 4°C then chased at 37°C for 30 min. The intensity of BODIPY FL-labeled sphingomyelin in cells was analyzed by (C) confocal fluorescence imaging and by (D) quantitative FACS (n = 3 biological replicates, ***p<0.001). Scale bars: 20 μm. (E) SIRT1 KO mESCs have increased levels of endogenous sphingomyelin. WT and SIRT1 KO mESCs cultured in ESGRO medium were extracted and total levels of endogenous sphingomyelin were determined in extracts by an enzyme-coupled colorimetric assay as described in Materials and methods (n = 3 biological replicates, *p<0.05). (F–H) Deletion of SIRT1 in E14 mESCs leads to accumulation of sphingomyelin. (F) SIRT1 was deleted in E14 mESC line using CRISPR/cas9 mediated gene editing technology and (G) relative levels of BODIPY FL-labeled sphingomyelins were imaged and (H) measured (n = 2 independent clones with three biological replicates for each clone, ***p<0.001). GC-01, pCRISPR-CG01 vector. Scale bars: 20 μm.

The online version of this article includes the following source data and figure supplement(s) for figure 1:

**Source data 1.** Numerical data for bar graphs in D, E, and H.
**Source data 2.** Uncut immunoblots in F.
**Figure supplement 1.** Deletion of SIRT1 in ESCs results in a dramatic accumulation of sphingomyelin.

mESCs (*Figure 1—figure supplement 1E*), suggesting that accumulation of sphingolipids in SIRT1 KO mESCs is specific to sphingomyelin.

SIRT1 KO mESCs in above analyses were previously generated in R1 mES cell line using traditional gene targeting technology (*McBurney et al., 2003*). To further confirm our observation that SIRT1 deficiency in mESCs induces accumulation of sphingomyelin, we deleted *Sirt1* gene in another widely used full pluripotent mES cell line, E14 cells (*Wakayama et al., 1999*), by CRISPR/Cas9-mediated gene editing technology (*Figure 1F*). Consistent with observations in SIRT1 KO mESCs, these SIRT1 KO E14 mESC clones also had an enhanced staining of BODIPY FL-C$_5$-sphingomyelin when analyzed by confocal fluorescence imaging (*Figure 1G*) and by quantitative FACS analysis (*Figure 1H*). Taken together, our results indicate that deletion of SIRT1 in ESCs results in accumulation of sphingomyelin in independent ES cell lines.

## Deletion of SIRT1 induces accumulation of sphingomyelin through SMPDL3B

Cellular levels of sphingomyelin are regulated by a tight balance between their synthesis and breakdown, which are mediated by activities of sphingomyelin synthases (SGMSs) and sphingomyelin phosphodiesterases (SMPDs), respectively (*Figure 1—figure supplement 1A*). Many of these enzymes were highly expressed in mESCs (*Figure 2—figure supplement 1A and B*). To better understand how SIRT1 deficiency in mESCs leads to sphingomyelin accumulation, we surveyed the expression levels of these enzymes in WT and SIRT1 KO mESCs. SIRT1 KO mESCs had significantly reduced expression of a sphingomyelin synthase *Sgms2* (*Figure 2A* and *Figure 2—figure supplement 1B*), and a dramatic reduction of a sphingomyelin phosphodiesterase SMPDL3B, one of the most highly expressed SMPDs in mESCs (*Figure 2—figure supplement 1A*), in both ESGRO and M10 media (*Figure 2A–C*). Since sphingomyelin was accumulated in SIRT1 KO mESCs, we focused on the reduction of *Smpdl3b*. As shown in *Figure 2D*, the reduced expression of SMPDL3B in SIRT1 KO mESCs was coupled with a decreased rate to clear away preloaded BODIPY FL-C$_5$-sphingomyelin in a time-lapse video analysis, suggesting that reduction of SMPDL3B-mediated sphingomyelin degradation may be responsible for accumulation of sphingomyelin observed in SIRT1 KO mESCs.

To test this possibility, we manipulated the levels of SMPDL3B in WT and SIRT1 KO mESCs and analyzed their impacts on cellular levels of sphingomyelin. Stable overexpression of SMPDL3B significantly reduced accumulation of BODIPY FL-C$_5$-sphingomyelin in SIRT1 KO but not WT mESCs when cells were cultured in serum-containing M10 medium (*Figure 3A–C*). In serum-free ESGRO medium, overexpression of SMPDL3B reduced accumulation of BODIPY FL-C$_5$-sphingomyelin and endogenous sphingomyelin in both WT and SIRT1 KO mESCs (*Figure 3—figure supplement 1A–C*). Moreover, the ability of SMPDL3B to reduce cellular levels of sphingomyelin is dependent on its enzymatic activity, as a catalytic inactive mutant of this enzyme, SMPDL3B H135A (*Heinz et al., 2015*; *Mitrofanova et al., 2019*), failed to decrease the levels of BODIPY FL-C$_5$-sphingomyelin and endogenous sphingomyelin in WT and SIRT1 KO mESCs (*Figure 3—figure supplement 1D*, *Figure 3D–F*). These results indicate that SMPDL3B is capable of removing sphingomyelin in mESCs, particularly in SIRT1 KO mESCs. Conversely, shRNA-mediated stable knockdown of SMPDL3B (*Figure 3G*) enhanced the accumulation of BODIPY FL-C$_5$-sphingomyelin (*Figure 3H and I*) and endogenous sphingomyelin (*Figure 3J*) in WT mESCs but not further in SIRT1 KO mESCs, indicating that accumulation of sphingomyelin observed in SIRT1 KO mESCs is primarily due to reduced expression of SMPDL3B.

## SIRT1 promotes transcription of *Smpdl3b* through c-Myc

As an NAD$^+$-dependent protein deacetylase that deacetylates histones, transcription factors, cofactors, as well as splicing factors, SIRT1 has been shown to modulate gene expression at multiple levels. We confirmed that SIRT1 indeed regulates the expression of *Smpdl3b* and sphingomyelin degradation through its catalytic activity, as a SIRT1 catalytic inactive mutant (H355Y, HY) failed to rescue defective *Smpdl3b* expression and reduce BODIPY FL-C$_5$-sphingomyelin accumulation in SIRT1 KO mESCs compared to WT SIRT1 protein (*Figure 4*). When interrogated each step along the expression of *Smpdl3b* gene in SIRT1 KO mESCs, we found that qPCR primers designed to amplify different segments of mature *Smpdl3b* mRNA all detected a reduced abundance of the full-length mature *Smpdl3b* mRNA upon SIRT1 deletion in mESCs (*Figure 5—figure supplement 1A and B*).

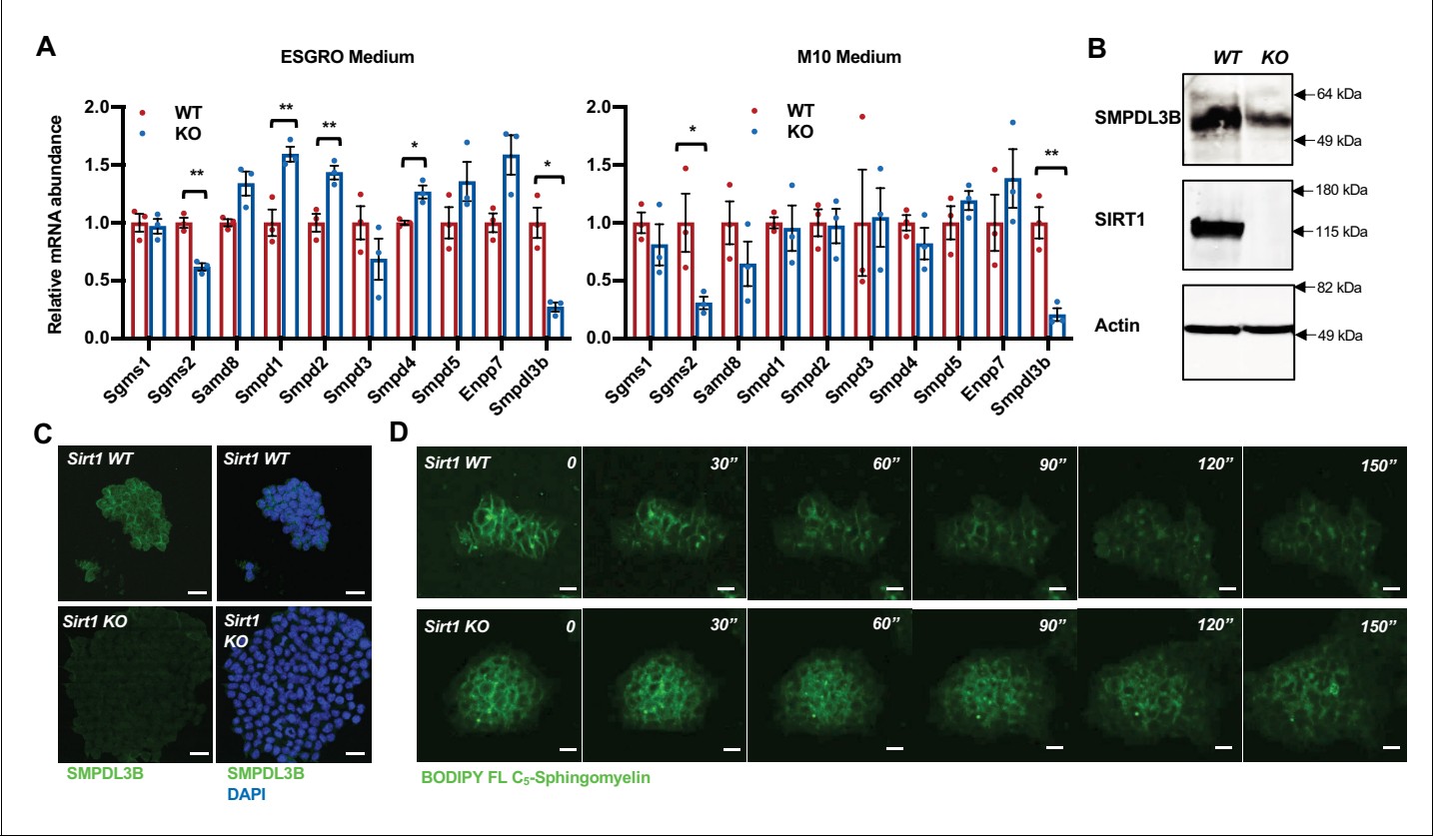

**Figure 2.** SIRT1-deficient mESCs have reduced expression of SMPDL3B and sphingomyelin degradation. (**A**) SIRT1 KO mESCs have reduced mRNA levels of *Smpdl3b*. WT and SIRT1 KO mESCs were cultured in either ESGRO medium or M10 medium. The mRNA levels of indicated enzymes involved in sphingomyelin synthesis (Sgms) and degradation (Smpd) were analyzed by qPCR (n = 3 biological replicates, *p<0.05, **p<0.01). (**B–C**) SIRT1 KO mESCs have reduced protein levels of SMPDL3B. The protein levels of SMPDL3B were analyzed by (**B**) immunoblotting and (**C**) immuno-fluorescence staining. Scale bars: 20 μm. (**D**) SIRT1 KO mESCs have reduced degradation of sphingomyelin. WT and SIRT1 KO mESCs were preloaded with BODIPY FL-C$_5$ sphingomyelin for 30 min at 4°C, then incubated with BODIPY FL-C$_5$ sphingomyelin-free medium at 37°C. The dynamic of BODIPY FL-sphingomyelin was monitored for additional 12 hr at 37°C. WT and SIRT1 KO mESC clones that have comparable preloaded levels of BODIPY FL-C$_5$ sphingomyelin were shown. Scale bars: 20 μm.

The online version of this article includes the following source data and figure supplement(s) for figure 2:

**Source data 1.** Numerical data for bar graphs in A.
**Source data 2.** Uncut immunoblots in B.
**Figure supplement 1.** Expression of sphingolipid synthesis and degrading enzymes in WT and SIRT1 KO mESCs.
**Figure supplement 1—source data 1.** Numerical data for bar graph in A.

Moreover, the abundance of *Smpdl3b* mRNA was reduced in both nuclear and cytosolic fractions in SIRT1 KO mESCs (*Figure 5—figure supplement 1C*). Northern blotting analysis using random probes generated from the full-length *Smpdl3b* cDNA further showed that deletion of SIRT1 in mESCs reduces the abundance of the full-length *Smpdl3b* mRNA without detectable accumulation of other minor isoforms (*Figure 5—figure supplement 1D*). Finally, RNA-seq analysis of total Ribo-minus RNA (total RNA after depletion of ribosomal RNAs) revealed that the abundance of RNA species from both exonic and intronic regions of *Smpdl3b* gene were reduced in SIRT1 KO mESCs (*Figure 5—figure supplement 1E* and *Supplementary file 3*), and no defective splicing of *Smpdl3b* pre-mRNA was detected in these cells (not shown). All these observations strongly suggest that the reduction of SMDPL3B expression in SIRT1 KO mESCs is due to defective transcription of *Smpdl3b* gene. In support of this notion, SIRT1 KO mESCs had a drastic depletion of Pol II near the TSS of *Smpdl3b* gene, along with decreased deposition of an activation mark H3K4me3 yet increased deposition of a repression mark H3K27me3 (*Figure 5A*), indicative of a strong attenuation of transcriptional activation of *Smpdl3b* gene in SIRT1 KO mESCs.

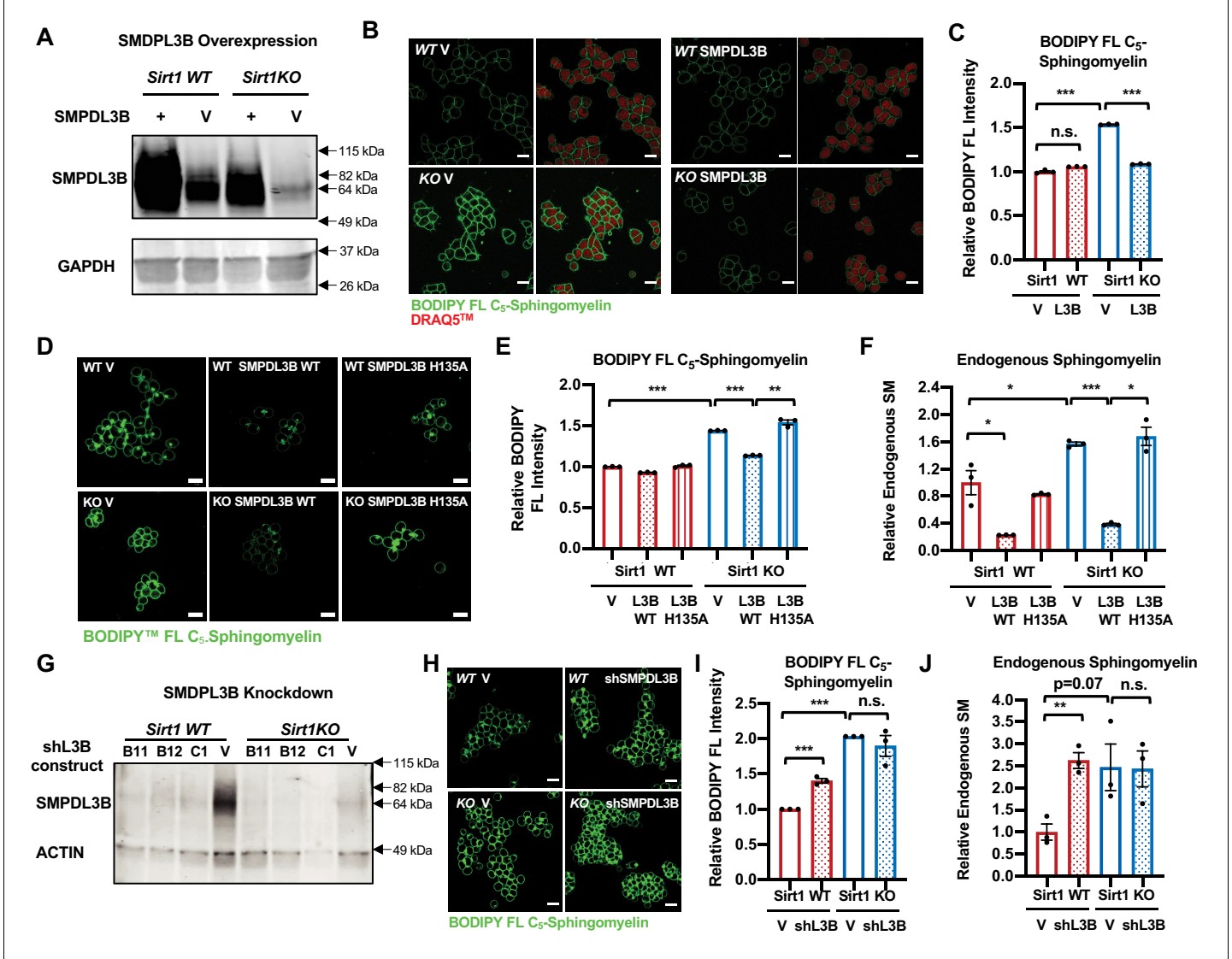

**Figure 3.** SMPDL3B directly controls the sphingomyelin contents in mESCs. (**A**) Overexpression of SMPDL3B in mESCs. WT and SIRT1 KO mESCs were infected with lentiviral particles containing empty vector (V) or a construct expressing SMPDL3B. The expression of SMPDL3B was analyzed by immuno-blotting. (**B–C**) Overexpression of SMPDL3B reduces sphingomyelin levels in mESCs cultured in M10 medium. The cellular levels of sphingomyelin in WT and SIRT1 KO mESCs with or without overexpression of SMPDL3B were analyzed by (**B**) BODIPY FL-sphingomyelin confocal imaging, and (**C**) FACS assay (n = 3 biological replicates, *p<0.05, **p<0.01). Scale bars in (**B**): 20 μm. L3B in (**C**): SMPDL3B. (**D–F**) Overexpression of WT but not a catalytic inactive mutant SMPDL3B reduces sphingomyelin levels in mESCs cultured in in M10 medium. WT and SIRT1 KO mESCs transfected with an empty vector (V), a construct expressing WT SMPDL3B protein (SMPDL3B WT), or a construct expressing a catalytic inactive mutant SMPDL3B protein (SMPL3B H135A). The cellular levels of sphingomyelin in these transfected cells were analyzed by (**D**) BODIPY FL-sphingomyelin staining, (**E**) BODIPY FL-sphingomyelin FACS assay, or (**F**) an enzyme-coupled colorimetric assay for endogenous sphingomyelin. (n = 3 biological replicates, *p<0.05, **p<0.01, ***p<0.001). (**G**) Stable knockdown of the expression of SMPDL3B in mESCs. WT and SIRT1 KO mESCs were infected with lentiviral particles containing empty vector (V) or shRNA constructs for SMPDL3B (B11, B12, C1). The expression of SMPDL3B were analyzed by immuno-blotting. shL3B: shRNAs against SMPDL3B. (**H–J**) Knocking down SMPDL3B increases sphingomyelin levels in WT mESCs but not significantly further in SIRT1 KO mESCs in ESGRO medium. The cellular levels of sphingomyelin in WT and SIRT1 KO mESCs with or without stable knockdown of SMPDL3B were analyzed by (**H**) BODIPY FL-sphingomyelin confocal imaging, (**I**) BODIPY FL-sphingomyelin FACS assay, or (**J**) an enzyme-coupled colorimetric assay for endogenous sphingomyelin (n = 3 biological replicates, **p<0.01, ***p<0.001). Scale bars: 20 μm.

The online version of this article includes the following source data and figure supplement(s) for figure 3:

**Source data 1.** Numerical data for bar graphs in C, E, F, I, and J.

**Source data 2.** Uncut immunoblots in A and G.

**Figure supplement 1.** SMPDL3B directly controls the sphingomyelin contents in mESCs.

**Figure supplement 1—source data 1.** Numerical data for bar graphs in B and C.

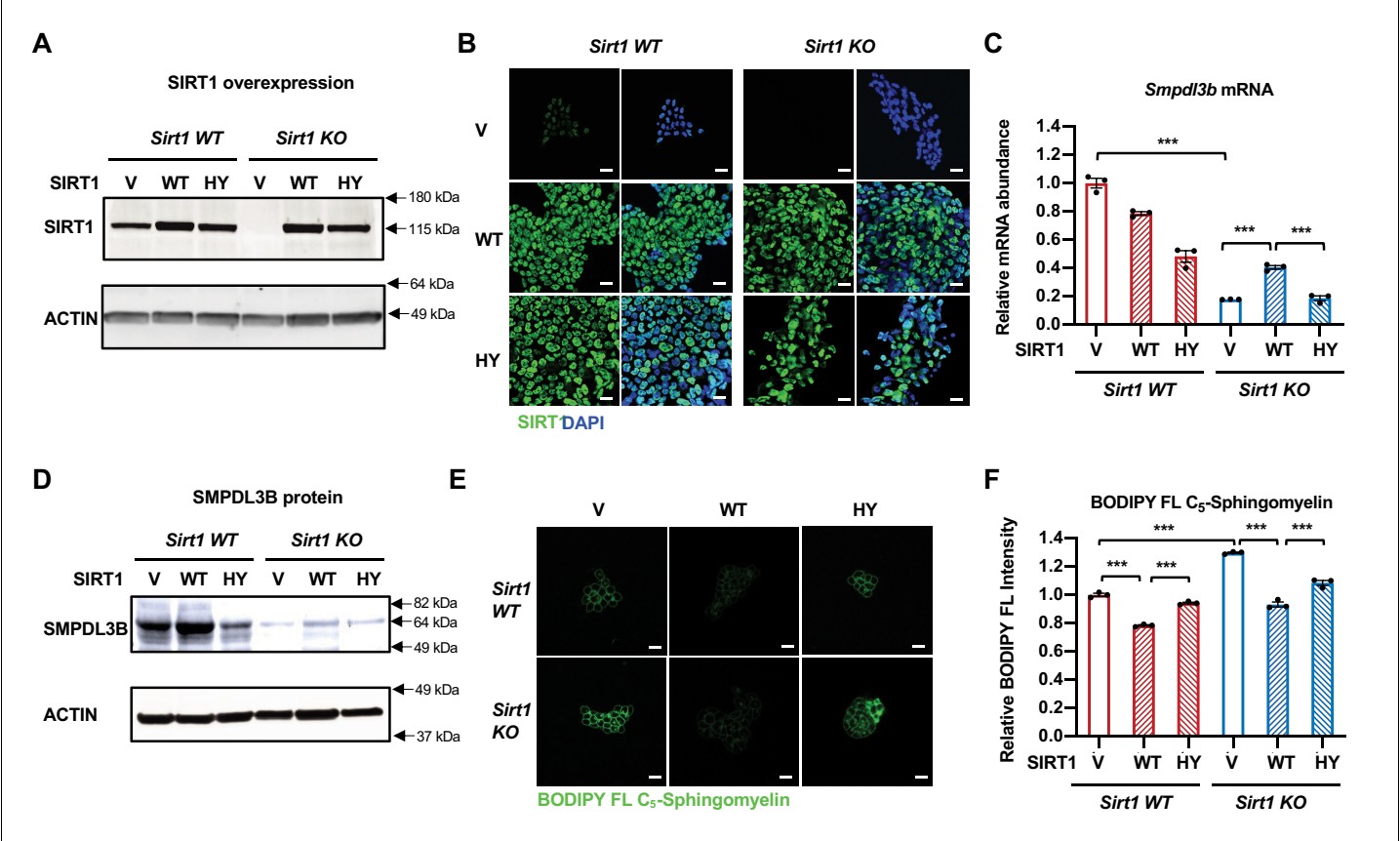

**Figure 4.** Expression of WT but not catalytically inactive SIRT1 partially rescues the sphingomyelin defect in SIRT1 KO mESCs. (**A–B**) SIRT1 protein levels in indicated mESCs. WT and SIRT1 KO mESCs were infected with lentiviral particles containing empty vector (PLenti-III-EF1α) or constructs expressing WT or a catalytically inactive mutant SIRT1 (H355Y, HY). The expression of SIRT1 was analyzed by either (**A**) immunoblotting or (**B**) immunofluorescence staining. Bars in B: 20 μm. (**C–D**) Expression of the HY mutant SIRT1 represses the expression of Smpdl3b in WT mESCs, whereas expression of WT but not the HY mutant SIRT1 increases the expression of Smpdl3b in SIRT1 KO mESCs. The expression of SMPDL3B was analyzed by either (**C**) qPCR or (**D**) immunoblotting. (n = 3 biological replicates, ***p<0.001). (**E–F**) Expression of WT but not HY mutant SIRT1 significantly reduces the sphingomyelin levels in both WT and SIRT1 KO mESCs. Indicated WT and SIRT1 KO mESCs cultured in ESGRO medium were labeled with BODIPY FL-labeled sphingomyelin for 30 min at 4 ˚C then incubated at 37˚C for 30 min. The intensity of BODIPY FL-labeled sphingomyelin in cells were analyzed by (**E**) confocal fluorescence imaging and by (**F**) quantitative FACS (n = 3 biological replicates, ***p<0.001). Bars in E: 20 μm.

The online version of this article includes the following source data for figure 4:

**Source data 1.** Numerical data for bar graphs in C and F.
**Source data 2.** Uncut immunoblots in A and D.

Sequence analysis of the TSS region revealed multiple potential transcription factors (TFs) that may target *Smpdl3b* gene, including two known SIRT1 deacetylation substrates, c-Myc and N-Myc (*Tang et al., 2017*; *Menssen et al., 2012*; *Figure 5B*). To determine the promoter region(s) and associated TF(s) that are responsible for the transcription suppression of *Smpdl3b* gene in SIRT1 KO mESCs, we designed small gRNAs (sgRNAs) to target different potential TF loci along the *Smpdl3b* promoter (*Figure 5—figure supplement 2A*, top), then analyzed their impacts on the expression of *Smpdl3b* after transfecting into WT and SIRT1 KO mESCs generated from a mouse ES cell line stably expressing a dox-inducible dCas9 and BirA-V5 (dCas9 mESCs, *Figure 5—figure supplement 2B*; *Liu et al., 2017*). It has been demonstrated that transfected sgRNAs in these cells are able to guide the deactivated Cas9 (dCas9) to bind to their targeting loci without further cleavage, resulting in altered transcription of the target gene (*Liu et al., 2017*). Compared to control Gal4 sgRNA and other sgRNAs, a sgRNA targeting a locus near 528 bp downstream of the TSS of *Smpdl3b* gene rescued the defective expression of *Smpdl3b* mRNA in SIRT1 KO dCas9mESCs (*Figure 5C*). Further

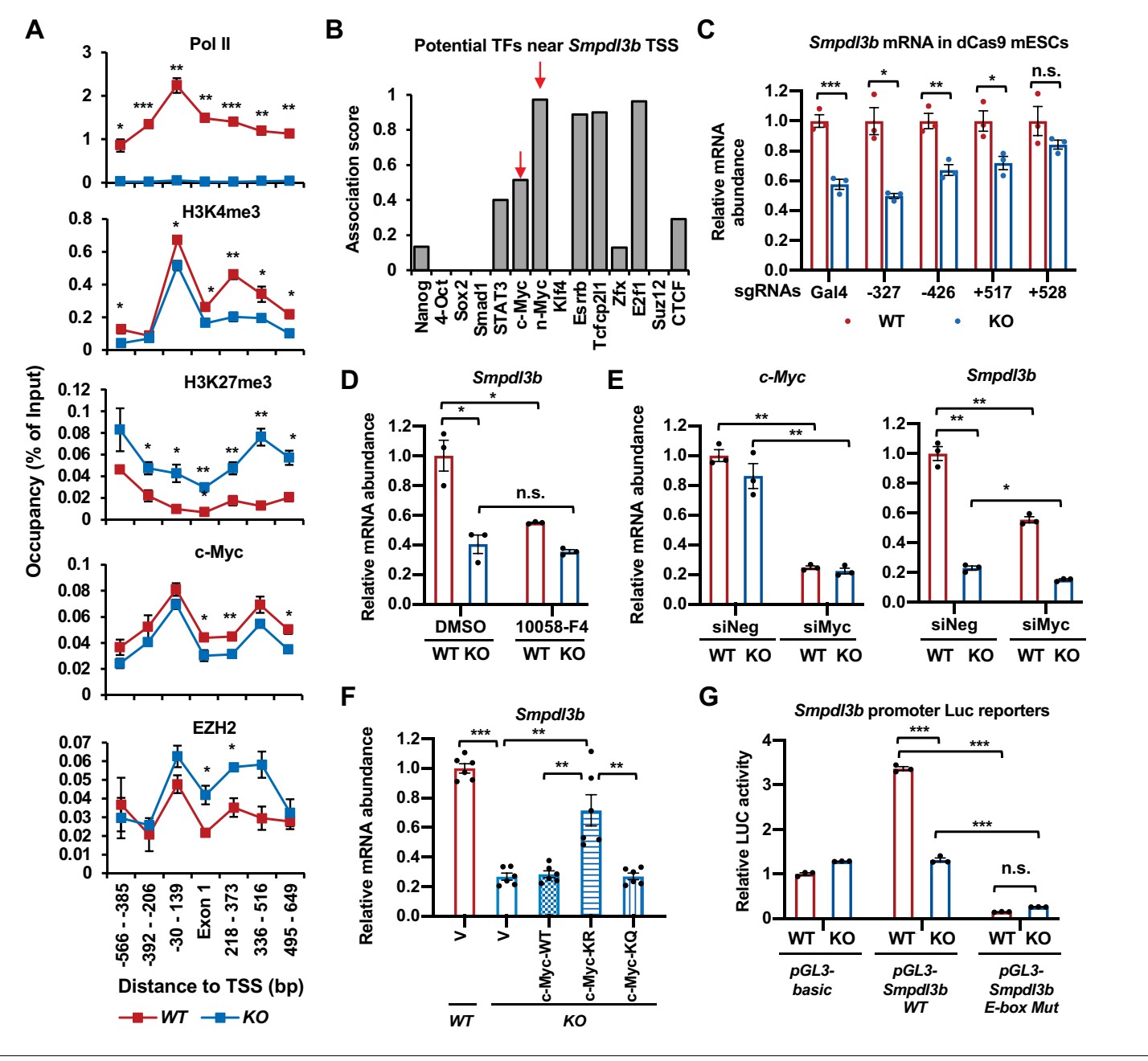

**Figure 5.** SIRT1 promotes the transcription of *Smpdl3b* through c-Myc in mESCs. (**A**) SIRT1 KO mESCs have reduced transcription of *Smpdl3b*. WT and SIRT1 KO mESCs cultured in ESGRO medium were crosslinked and subjected for ChIP-qPCR profiling of PolII, c-Myc, EZH2, and indicated chromatin activation or repression marks near the TSS region of *Smpdl3b* gene (n = 4 biological replicates, *p<0.05, **p<0.01, ***p<0.001). (**B**) Association scores of potential transcription factors (TFs) near the TSS of *Smpdl3b* gene. The association scores of indicated TFs were obtained from a published dataset (*Evans et al., 2014*). A higher score is suggestive of a higher chance of *Smpdl3b* gene being targeted by the potential TF. (**C**) A guide RNA (gRNA) targeting the +528 locus at the TSS region of *Smpdl3b* gene rescues the expression of this gene. sgRNAs targeting indicated loci near the TSS region of *Smpdl3b* gene were transfected into WT and SIRT1 KO mESCs stably expressing a dox-inducible dCas9 and BirA-V5 (dCas9 mESCs). The mRNA levels of *Smpdl3b* were analyzed by qPCR (n = 3 biological replicates, *p<0.05, **p<0.01, ***p<0.001). (**D**) Inhibition of c-Myc activity reduces the expression of *Smpdl3b* gene in mESCs. WT and SIRT1 KO mESCs were treated with DMSO or 10 mM 10058-F4 for 48 hr. The mRNA levels of *Smpdl3b* were analyzed by qPCR (n = 3 biological replicates, *p<0.05, **p<0.01, ***p<0.001). (**E**) Knocking down c-Myc significantly reduces the expression of *Smpdl3b* gene in mESCs. WT and SIRT1 KO mESCs were transfected with siRNAs against c-Myc for 48 hr. The mRNA levels of *c-Myc* and *Smpdl3b* were analyzed by qPCR (n = 3 biological replicates, *p<0.05, **p<0.01). (**F**) Overexpression of the KR mutant of c-Myc partially reduces the expression of *Smpdl3b* gene in SIRT1 KO mESCs. The mRNA levels of *Smpdl3b* in indicated mESCs were analyzed by qPCR (n = 6 biological replicates, **p<0.01, ***p<0.001). (**G**) Mutation of the c-Myc binding E-box element on the promoter of *Smpdl3b* gene abolishes the expression of *Smpdl3b* luciferase

*Figure 5 continued on next page*

*Figure 5 continued*

reporter in mESCs. Luciferase reporters containing the basic vector (pGL3-basic), the WT promoter of *Smpdl3b* gene, or a promoter of *Smpdl3b* gene with a mutant E-box were transfected into WT and SIRT1 KO mESCs, and the luciferase activities were measured as described in Materials and methods (n = 3 biological replicates, ***p<0.001).

The online version of this article includes the following source data and figure supplement(s) for figure 5:

**Source data 1.** Numerical data for graphs in A, B, C, D, E, F, and G.

**Figure supplement 1.** Deletion of SIRT1 reduces transcription of *Smpdl3b* gene.

**Figure supplement 1—source data 1.** Numerical data for bar graphs in B and C.

**Figure supplement 2.** SIRT1 promotes the transcription of *Smpdl3b* through c-Myc but not EZH2 in mESCs.

**Figure supplement 2—source data 1.** Numerical data for bar graphs in D, E, and G.

**Figure supplement 2—source data 2.** Uncut immunoblots in B.

bioinformatic analysis showed that this locus is overlapped with previously mapped binding regions of two TFs, c-Myc and EZH2 (*Figure 5—figure supplement 2A*, bottom). c-Myc is a known SIRT1 deacetylation substrate, and deacetylation of c-Myc by SIRT1 has been reported to increase its stability and activity (*Menssen et al., 2012*). We have previously shown that c-Myc is hyperacetylated but unstable in SIRT1 KO mESCs, which reduces its binding to target promoters thereby decreasing their transcription (*Tang et al., 2017*). The association of c-Myc protein to the promoter of *Smpdl3b* gene was indeed significantly reduced in SIRT1 KO mESCs by a ChIP-qPCR assay (*Figure 5A*, c-Myc). Moreover, inhibition of c-Myc activity by 10058-F4 (*Yin et al., 2003*; *Figure 5D*) or knocking down c-Myc with siRNAs (*Figure 5E*) significantly reduced the mRNA abundance of *Smpdl3b* in WT mESCs but not or to a less extend in SIRT1 KO mESCs, indicating that c-Myc is a key transcription factor in SIRT1-mediated regulation of *Smpdl3b*. Furthermore, SIRT1 promoted the transcription of *Smpdl3b* in part through deacetylating c-Myc, as a c-Myc mutant with its major acetylation site mutated to R to mimic deacetylated c-Myc (K323R, KR), partially rescued the expression of *Smpdl3b* in SIRT1 KO mESCs compared to empty vector (V), WT c-Myc, and a c-Myc mutant with its major acetylation site mutated Q to mimic acetylated c-Myc (K323Q, KQ) (*Figure 5—figure supplement 2C* and *Figure 5F*). Finally, a *Smpdl3b* promoter luciferase reporter containing a mutant c-Myc binding site (E-box) displayed a dramatically reduced activity in mESCs compared to a WT *Smpdl3b* promoter luciferase reporter (*Figure 5G*, pGL3-Smpdl3b E-box Mut vs pGL3-Smpdl3b WT in WT mESCs). Additionally, this mutant luciferase reporter had a comparable low activity in SIRT1 KO mESCs vs WT mESCs, further suggesting that the differential expression levels of *Smpdl3b* in WT and SIRT1 KO mESCs is mediated by c-Myc.

EZH2, an H3K27me3 methyltransferase and the functional enzymatic component of the Polycomb Repressive Complex 2 (PRC2), is also a deacetylation substrate of SIRT1 (*Wan et al., 2015*). Deacetylation of K348 of EZH2 by SIRT1 has been reported to reduce its stability and activity (*Wan et al., 2015*), which is consistent with our current observation that deletion of SIRT1 in mESCs significantly increased the occupancy of EZH2 and H3K27me3 on the promoter of *Smpdl3b* gene (*Figure 5A*, EZH2, H3K27me3). However, in contrast to c-Myc, neither inhibition of EZH2 activity by its inhibitor EPZ6438 (*Figure 5—figure supplement 2D*) nor knockdown of EZH2 by siRNAs (*Figure 5—figure supplement 2E–G*) significantly affected the expression of *Smpdl3b* in mESCs, particularly in SIRT1 KO mESCs, indicating that blocking EZH2-catalyzed H3K27 trimethylation alone is not sufficient to rescue SIRT1 deficiency-induced transcriptional suppression of *Smpdl3b* in mESCs. Collectively, our data demonstrate that SIRT1 activates the expression of *Smpdl3b* in mESCs primarily through deacetylation of c-Myc.

## SIRT1-regulated sphingolipid metabolism affects in vitro neural differentiation

SMPDL3B-catalyzed sphingomyelin degradation has been shown to reduce plasma membrane fluidity (*Heinz et al., 2015*). As expected from their reduced expression of SMPDL3B, SIRT1 KO mESCs had a reduced fraction of ordered structures (thereby increased membrane fluidity) compared to WT mESCs when probed with Di-4-ANEPPDHQ, an electrical potential sensitive fluorescent dye for detection of microdomains and (dis)ordered membrane in live cells (*Figure 6A*, vehicle). Treatment with methyl-β-cyclodextrin (MβCD), a cholesterol-extracting agent, further decreased the membrane

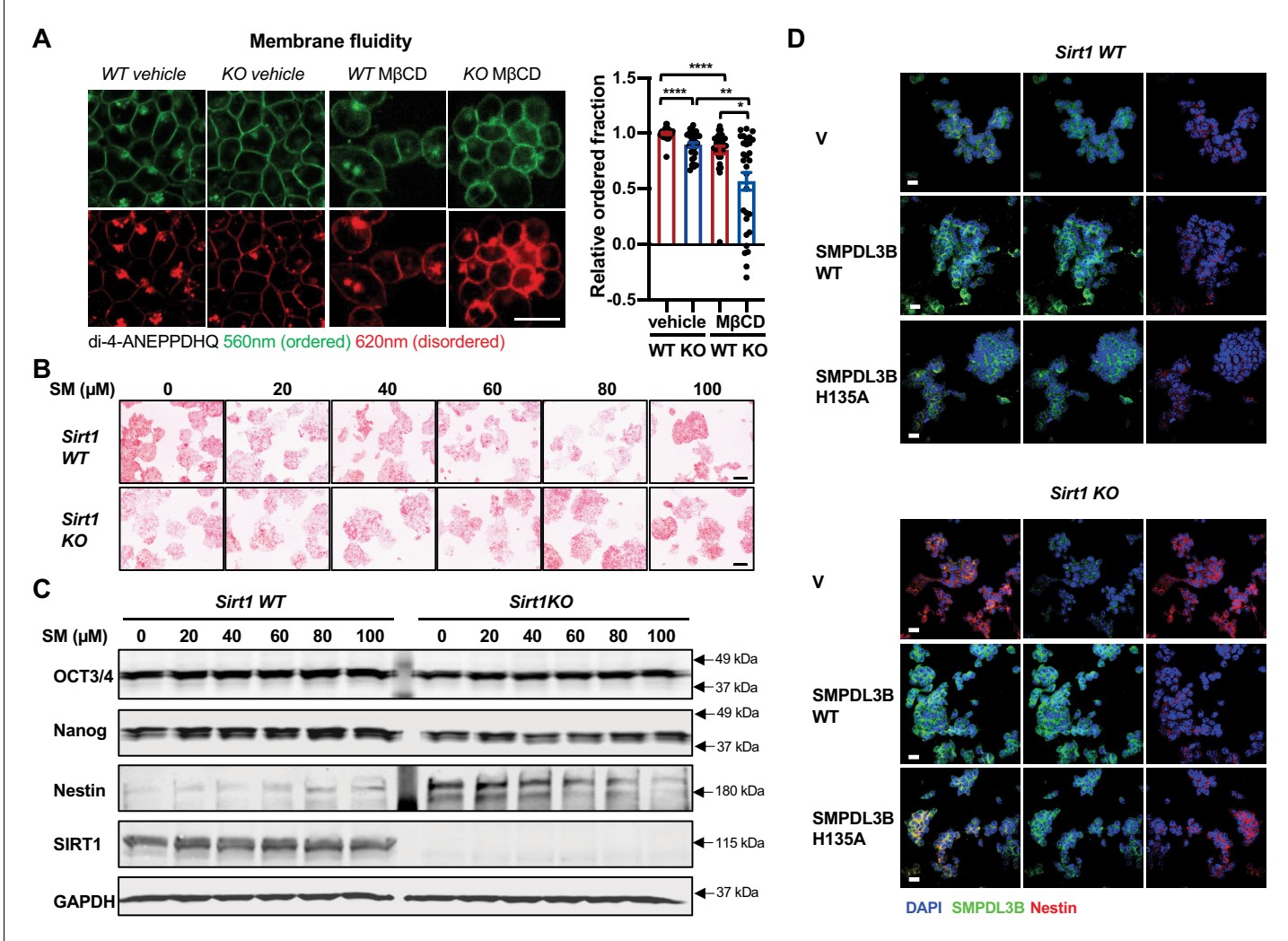

**Figure 6.** Sphingomyelin accumulation increases membrane fluidity and induces expression of Nestin in SIRT1 KO mESCs. (A) SIRT1 KO mESCs have an increased membrane fluidity. WT and SIRT1 KO mESCs cultured in ESGRO medium were preincubated with or without 2.5 mM MβCD for 1 hr, then stained with 5 μM di-4-ANEPPDHQ for at least 30 min. The relative ordered fraction in each group was analyzed as described in Materials and methods (n = 30 clones/group, *p<0.05, **p<0.01, ***p<0.001, ****p<0.0001). Bars: 10 μm. (B–C) Exogeneous sphingomyelin treatment increases Nestin but not pluripotency markers in mESCs. WT and SIRT1 KO mESCs were treated with indicated concentrations of sphingomyelin (SM) in ESGRO medium for 48 hr. (B) The intensity of AP was analyzed as described in Materials and methods. Bars: 100 μm. (C) The protein abundance of pluripotency marker OCT3/4, Nanog and neuroepithelial stem cell marker Nestin in WT and Sirt1 KO mESCs were determined by immunoblotting. (D) Overexpression of WT but not a catalytic inactive mutant SMPDL3B reduces the expression of Nestin in SIRT1 KO mESCs. WT and SIRT1 KO mESCs transfected with an empty vector (V), a construct expressing WT SMPDL3B protein (SMPDL3B WT), or a construct expressing a catalytic inactive mutant SMPDL3B protein (SMPL3B H135A) were stained for SMPDL3B and Nestin. Scale bars: 20 μm.

The online version of this article includes the following source data and figure supplement(s) for figure 6:

**Source data 1.** Numerical data for bar graphs in A.

**Source data 2.** Uncut immunoblots in C.

**Figure supplement 1.** SIRT1 deficiency in mESCs significantly reduces the expression of genes involved in signaling pathways.

order, particularly in SIRT1 KO mESCs (*Figure 6A*, MβCD). In line with this observation, pathways involved in cell surface receptor signaling pathway and intracellular signal transduction were among the most significantly disrupted Gene Ontology (GO) Biological Process (GO BP) in SIRT1 KO mESCs when compared with WT mESCs in our Ribo-minus RNA-seq dataset (*Figure 6—figure supplement 1*).

Sphingolipids are bioactive lipids important for stem cell survival and differentiation (*Bieberich, 2008*; *Wang et al., 2018*). Since we previously observed that SIRT1 KO mESCs have a compromised pluripotency (*Tang et al., 2017*), we investigated whether sphingomyelin accumulation in SIRT1 KO mESCs is responsible for their reduced pluripotency. Compared to WT mESCs, SIRT1 KO mESCs had a reduced staining intensity of Alkaline Phosphatase (AP), a marker of undifferentiated ESCs (*Figure 6B*), along with decreased expression of OCT3/4 and Nanog, two pluripotent stem cell markers, and increased expression of Nestin, a neuroectodermal stem cell (NSC) marker (*Figure 6C*). However, sphingomyelin treatment did not consistently affect the AP staining intensity (*Figure 6B*) nor induced any significantly changes on the expression of pluripotency markers in either WT or SIRT1 KO mESCs (*Figure 6C*, OCT3/4 and Nanog). In contrast, sphingomyelin dose-dependently increased the expression of Nestin, a NSC marker that was induced in SIRT1 KO mESCs, but not in WT mESCs (*Figure 6C*, Nestin). Moreover, overexpression of WT but not H135A mutant SMPDL3B reduced the expression of Nestin in SIRT1 KO mESCs (*Figure 6D*), suggesting that SMPDL3B deficiency-resulted sphingomyelin accumulation may interfere neural differentiation in SIRT1 KO mESCs instead of their pluripotency.

Consistent with this observation, during a 4-week in vitro neural differentiation of mESCs (*Figure 7A*; *Ying et al., 2003*; *Abranches et al., 2009*), the mRNA levels of *Sirt1* and *Smpdl3b* were significantly reduced in WT E14 mESCs, along with dramatic decrease of *Nanog* and *Oct4* and massive induction of several NSC and neural differentiation factors, such as *Sox3*, *Nestin*, *Notch3*, and *Tau* (*Figure 7B*, WT). However, both the reduction of pluripotency markers and the induction of NSC/neural differentiation factors were significantly blunted when SIRT1 was deleted in E14 mESCs (*Figure 7B*, KO), indicating that SIRT1 deficiency impairs in vitro neural differentiation of these cells. Further cellular and morphological analyses by immunofluorescence staining of progenitor and neuronal markers showed that progenitors and neurons differentiated from WT E14 mESCs have high expression of marker proteins and typical mature neuronal morphology, including elongated axons and dendrites, after 4-week differentiation (*Figure 7C*, WT). Progenitors and neurons differentiated from SIRT1 KO E14 mESCs, on the other hand, had low levels of these markers and lacked typical neuronal morphology (*Figure 7C*, KO). Additionally, SIRT1 KO mESCs also displayed defective neural differentiation after in vitro differentiation (*Figure 7D*), indicating that SIRT1 deficiency in mESCs impairs neural differentiation in vitro in a cell line independent manner.

To validate that defective neural differentiation of SIRT1 KO mESCs is related to their accumulation of sphingomyelin, we analyzed whether adding back SMPDL3B in these cells will rescue their neural differentiation defects. Morphologically, putting back SMPDL3B into SIRT1 KO mESCs increased neurons with elongated axons and dendrites (*Figure 8A and B*, SIRT1 KO SMPDL3B), which was associated with the increased fraction of cells positive of several neural markers when analyzed by FACS analysis (*Figure 8C*, SIRT1 KO SMPDL3B). Moreover, the expression of several progenitor and neuronal markers was also increased by adding back SMPDL3B in SIRT1 KO mESCs when analyzed by immunofluorescence staining (*Figure 8D*, SIRT1 KO SMPDL3B) or by qPCR (*Figure 8E*, SIRT1 KO SMPDL3B). Overexpression of SMPDL3B in WT mESCs, however, disrupted the expression of progenitor markers and morphology of neurons (*Figures 8B, C and D*, SIRT1 WT SMPDL3B), suggesting that a balanced sphingomyelin degradation is required to maintain normal neural differentiation. Conversely, in vitro neural differentiation of WT and SIRT1 KO mESCs with or without stable knockdown of SMPDL3B revealed that reduction of this enzyme disrupts neural differentiation in both WT and SIRT1 KO mESCs (*Figure 8F*), indicative the importance of this enzyme in normal neural differentiation. Finally, putting back WT SIRT1 protein into SIRT1 KO mESCs rescued expression of progenitor marker SOX1 and NSC marker Nestin as well as neuronal morphology after in vitro neural differentiation (*Figure 8G*, SIRT1 KO-WT). In contrast, putting back a catalytic inactive SIRT1 mutant failed to restore marker protein expression and/or neuronal morphology (*Figure 8G*, SIRT1 KO-HY), confirming that the neural differentiation defects observed in SIRT1 KO mESCs are primarily due to a lack of SIRT1 deacetylase activity. Taken together, our observations indicate that SIRT1-mediated transcription of *Smpdl3b* and sphingolipid degradation influence neural differentiation in vitro.

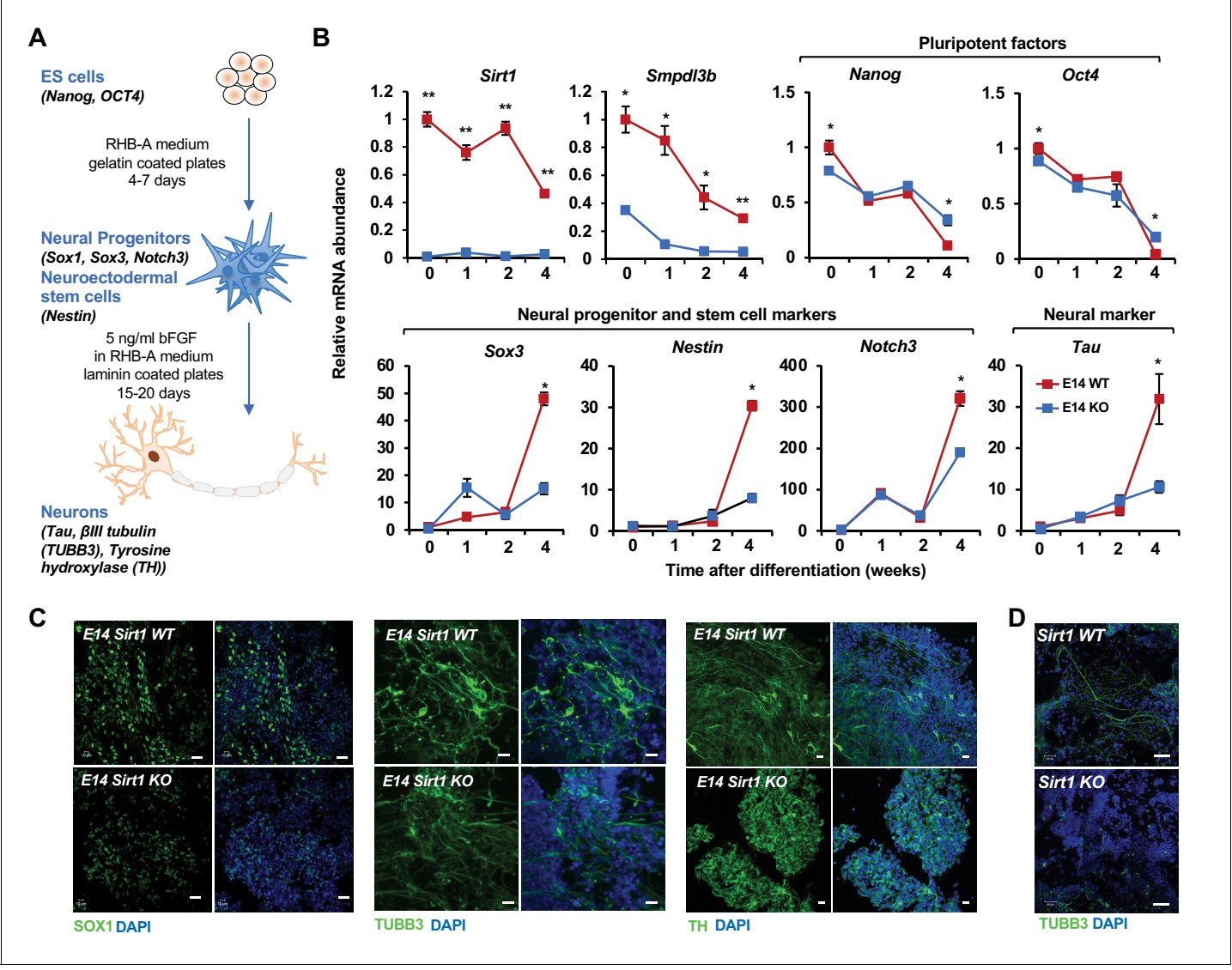

**Figure 7.** SIRT1 KO mESCs have an impaired neural differentiation in vitro. (A) A diagram of the in vitro neural differentiation system. (B) SIRT1 KO E14 mESCs are less responsive to in vitro neural differentiation than WT mESCs. The expression of indicated genes were analyzed by qPCR during 4 weeks of in vitro neural differentiation. Please note that deletion of SIRT1 resulted in reduction in both repression of pluripotent factors and induction of markers for neural progenitors/stem cells and neurons (n = 3 biological replicates, *p<0.05, **p<0.01). (C) SIRT1 KO E14 mESCs have reduced expression of neural differentiation markers and disordered neuronal morphology. WT and SIRT1 KO E14 mESCs after 4 weeks of in vitro neural differentiation were stained for a neural progenitor marker SOX1 (left panels) and neuronal markers beta III tubulin (TUBB3, middle panels) and TH (right panels). Scale bars: 20 μm. (D) SIRT1 KO mESCs have reduced expression of TUBB3 and mature neuronal morphology. WT and SIRT1 KO mESCs after 4 weeks of in vitro neural differentiation were stained for TUBB3. Scale bars: 50 μm.

The online version of this article includes the following source data for figure 7:

**Source data 1.** Numerical data for line graphs in B.

## Maternal HFD feeding induces accumulation of sphingomyelin and impairs neural development in SIRT1-deficient embryos

To assess the importance of sphingolipid metabolism in SIRT1-regulated neural differentiation in vivo, we investigated whether embryonic SIRT1 deficiency is associated with altered sphingomyelin accumulation and neural differentiation in mice. Consistent with our previous observations (*Tang et al., 2017*), systemic deletion of SIRT1 in C57BL/6J mice leads to intrauterine growth retardation when dams were fed on a regular chow diet (containing 4% fat) (*Figure 9A*, chow). The mRNA levels of *Smpdl3b* were significantly reduced in the brain of SIRT1 KO E18.5 embryos

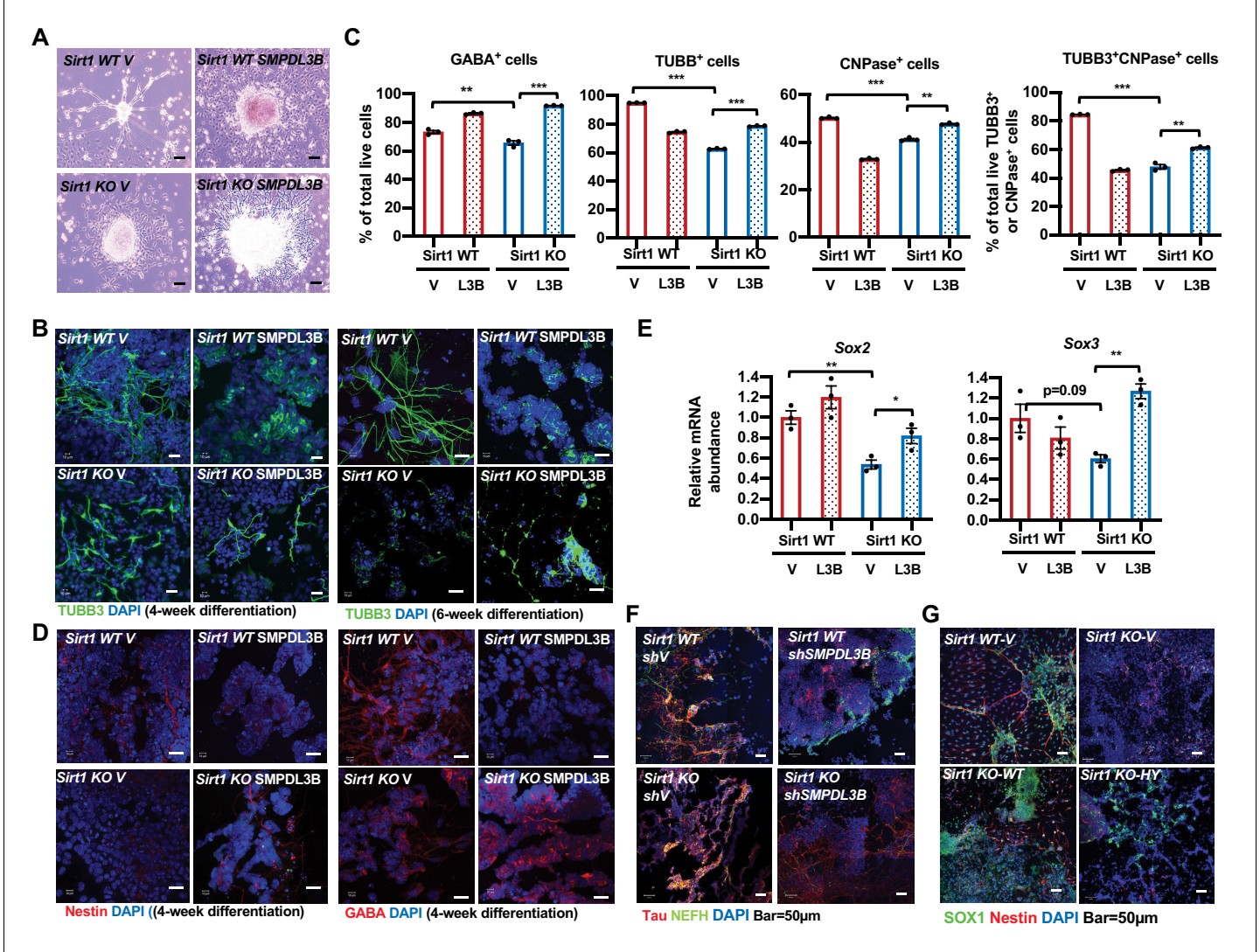

**Figure 8.** Reduced expression of SMPDL3B is partially responsible for impaired in vitro neural differentiation in SIRT1 KO mESCs. (A) Overexpression of SMPDL3B partially rescues gross neuronal morphology in in vitro differentiated SIRT1 KO mESCs. WT and SIRT1 KO mESCs stably infected with lentiviral particles containing empty vector (V) or constructs expressing SMPDL3B protein were subjected to 4 weeks of in vitro neural differentiation. The cell morphology was analyzed using regular light microscopy fixed with ZEISS AxioCamHR camera. Scale bars: 20 μm. (B) Overexpression of SMPDL3B partially rescues neuronal morphology in in vitro differentiated SIRT1 KO cells. WT and SIRT1 KO mESCs expressing vector (V) or SMPDL3B were differentiated for 4 weeks or 6 weeks. Six weeks of differentiation is for a better morphological analysis. The expression of TUBB3 and neuronal morphology were analyzed by immunofluorescence staining. Scale bars: 20 μm. (C) Overexpression of SMPDL3B partially increased the fraction of differentiated cells in in vitro differentiated SIRT1 KO cells. WT and SIRT1 KO mESCs expressing vector (V) or SMPDL3B were differentiated as in (A). The fraction of differentiated cells positive of indicated neural markers were quantified by FACS (n = 3 biological replicates, **p<0.01, ***p<0.001). (D) Overexpression of SMPDL3B partially rescues the expression of neural markers in in vitro differentiated SIRT1 KO cells. WT and SIRT1 KO mESCs expressing vector (V) or SMPDL3B were differentiated as in (A). The expression of indicated neural markers were analyzed by immunofluorescence staining. Scale bars: 20 μm. (E) Overexpression of SMPDL3B partially rescues the expression of neural progenitor markers in in vitro differentiated SIRT1 KO cells. WT and SIRT1 KO mESCs expressing vector (V) or SMPDL3B were differentiated as in (A). The expression of SOX2 and SOX3 were analyzed by qPCR (n = 3 biological replicates, *p<0.05, **p<0.01). (F) Knocking down SMPDL3B in WT mESCs impairs neural differentiation in vitro. WT and SIRT1 KO mESCs with or without stable knockdown of SMPDL3B were in vitro differentiated into neurons for 4 weeks. The expression of neural markers Tau and NEFH were analyzed by immunofluorescence staining. Scale bars: 50 μm. (G) WT but not a catalytic inactive SIRT1 rescues neural differentiation in vitro. WT and SIRT1 KO mESCs expressing vector (V), WT SIRT1, or a mutant SIRT1 lacking catalytic activity (HY) were in vitro differentiated into neurons for 4 weeks. The expression of neural markers SOX1 and Nestin were analyzed by immunofluorescence staining. Scale bars: 50 μm.

The online version of this article includes the following source data for figure 8:

**Source data 1.** Numerical data for bar graphs in C and E.

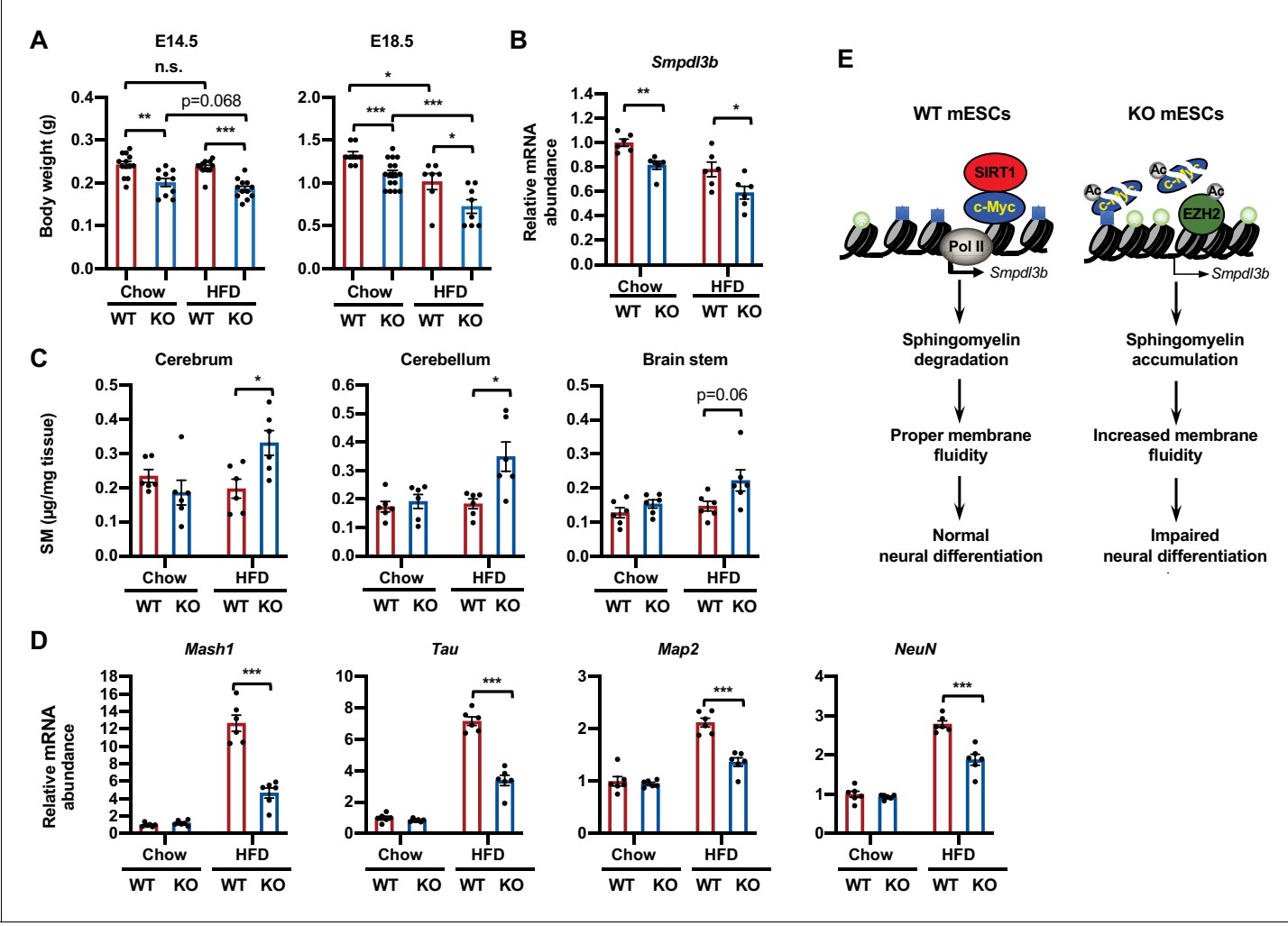

**Figure 9.** Maternal high-fat diet (HFD) feeding impairs neural development in SIRT1 deficient embryos. (**A**) Maternal HFD feeding reduces body weight of embryos. Maternal HFD feeding was performed 4-8 weeks before pregnancy (pre-feeding) as described in Materials and methods. Body weight of E14.5 and E18.5 embryos were measured (*p<0.05, **p<0.01, ***p<0.001). (**B**) SIRT1 KO embryos have reduced expression of *Smpdl3b* in brains. The mRNA levels of *Smpdl3b* in brain of E18.5 embryos from chow fed dams or HFD fed dams were analyzed by qPCR (n = 6 embryos, *p<0.05, **p<0.01). (**C**) Maternal HFD feeding induces sphingomyelin accumulation in brains of SIRT1 KO embryos. Maternal HFD feeding was performed 4-8 weeks before pregnancy (pre-feeding) as described in Materials and methods. Brains from E18.5 embryos were dissected into three parts and the endogenous sphingomyelins were extracted and measured (n = 6 embryos, *p<0.05). (**D**) Maternal HFD feeding induces defective expression of neural markers in brains of SIRT1 KO embryos. The mRNA levels of indicated neural markers in brain of E18.5 embryos from chow fed dams or HFD fed dams were analyzed by qPCR (n = 6 embryos, *p<0.05, **p<0.01, ***p<0.001). (**E**) SIRT1 regulates sphingomyelin degradation and neural differentiation of mESCs through c-Myc and EZH2. SIRT1 is highly expressed in mESCs cells, where it functions to promote association of c-Myc and recruitment of Pol II to activate transcription of *Smpdl3b* gene and subsequent sphingomyelin degradation. This action of SIRT1 is important for maintenance of a proper membrane fluidity for normal neural differentiation in response to nutritional/developmental cues. Deletion of SIRT1 causes hyperacetylation and instability of c-Myc, leading to Pol II depletion and transcriptional repression of *Smpdl3b*. SIRT1 deficiency-induced hyperacetylation and stabilization of EZH2 likely enforce this transcriptional suppression by adding H3K27me3 mark. This transcriptional repression of *Smpdl3b* is associated with accumulation of sphingomyelin, which increases membrane fluidity and impairs neural differentiation. Light blue squares: H3K4me3; Light green circles: H3K27me3; Ac: acetylation.

The online version of this article includes the following source data for figure 9:

**Source data 1.** Numerical data for bar graphs in A, B, C, and D.

(*Figure 9B*). However, these embryos did not display any detectable defects in brain sphingomyelin levels, nor in expression of a number of neural progenitor and neuron markers (*Figure 9C and D*, chow) despite reported developmental defects in other systems (*McBurney et al., 2003*; *Cheng et al., 2003*; *Wang et al., 2008*). Since HFD feeding has been shown to induce sphingolipid

biosynthesis and turnover of sphingolipids in multiple tissues (*Choi and Snider, 2015*), we tested whether maternal HFD feeding could induce sphingomyelin accumulation and disrupt neural development in SIRT1 KO embryos. Intriguingly, maternal feeding of an HFD diet containing 36% fat for 4–8 weeks before breeding significantly reduced intrauterine growth of embryos, particularly on SIRT1 KO embryos, at E18.5 (*Figure 9A*, HFD). Maternal HFD feeding also elevated sphingomyelin contents in all tested regions of brain in SIRT1 KO but not WT E18.5 embryos (*Figure 9C*, HFD). These maternal HFD feeding-induced gross and metabolic alterations were associated with reduced expression of many intermediate progenitor and mature neuron markers (*Figure 9D*, HFD) without significant changes on early stage neuroepithelial cell markers and oligodendrocyte markers (not shown), suggesting that SIRT1 deficiency-induced sphingomyelin accumulation specifically delays neuron maturation in mouse embryos.

## Discussion

Highly enriched in the nervous system, sphingolipids are important for the development and maintenance of the functional integrity of the nervous system (*van Echten-Deckert and Herget, 1758*; *Olsen and Færgeman, 2017*). Perturbations of the sphingolipid metabolism has been shown to rearrange the plasma membrane, resulting in development of various human diseases, particularly neurological diseases (*Piccinini et al., 2010*). However, despite these diverse biological functions, the transcriptional regulation of sphingolipid metabolism is largely unknown. In the present study, we show that cellular sphingomyelin degradation is under transcriptional control of SIRT1, an important cellular metabolic sensor. We provide evidence that the SIRT1-Myc axis is vital for transcriptional activation of SMPDL3B, a major GPI-anchored plasma-membrane-bound sphingomyelin phosphodiesterase in mESCs. This transcriptional regulation directly impacts cellular levels of sphingomyelin and membrane fluidity, and is important in regulation of neural differentiation in response to developmental signals (*Figure 9E*). Our findings therefore identify a unique genetic regulatory pathway for sphingolipid homeostasis. Given the high sensitivity of SIRT1 to nutritional and environmental perturbations (e.g. activation upon caloric restriction and repression after HFD feeding or during aging *Cantó and Auwerx, 2009*; *Imai, 2009*), our study further suggests that SIRT1-Myc-regulatetd sphingolipid degradation may be an important element in mediating reported environmental influence on sphingolipid metabolism (*Choi and Snider, 2015*; *Giusto et al., 1992*; *Lightle et al., 2000*; *Tacconi et al., 1991*). It will be of great interest to test this possibility in future studies.

As the most conserved mammalian NAD$^+$-dependent protein deacetylase, SIRT1 has a number of important functions in the brain, including regulation of late stage of neural development and protection against a number of neurodegenerative diseases (*Herskovits and Guarente, 2014*). In particular, SIRT1 has been shown to modulate the neural and glial specification of neural precursors (*Prozorovski et al., 2008*; *Kang et al., 2010*) and repress low glucose induced proliferation and neurogenesis of neural stem and progenitor cells (NSCs) in vitro (*Fusco et al., 2016*). Our observations in the present study demonstrate that SIRT1 is also a key metabolic regulator for the differentiation of neural progenitors/NSCs from ESCs. Our results show that SIRT1 is highly expressed in mESCs cells (*Figure 7B*), where it functions to promote c-Myc-mediated transcriptional activation of SMPDL3B and sphingomyelin degradation (*Figures 1–4*). This action of SIRT1 appears to have minimal impacts on the pluripotency of mESCs (*Figure 6B and C*), but instead is important for maintenance of a proper membrane fluidity for normal neural differentiation in response to nutritional/developmental cues (*Figure 9E*). Thus, by interacting with different protein factors, SIRT1 is important for neural differentiation and development at multiple stages.

How impaired degradation of sphingomyelin might influence the differentiation of ESCs remains unclear. Our observations that SMPDL3B deficiency in SIRT1 KO mESCs is associated with increase of membrane fluidity (*Figure 6A*) and that pathways involved in cell surface receptor signaling pathway are one of the most significantly downregulated biological processes in SIRT1 KO mESCs (*Figure 6—figure supplement 1*) suggest that impaired signaling transduction may underlie the impaired neural differentiation of these cells. This idea is consistent with the notion that sphingomyelin is important for formation of microdomains/lipid rafts on the plasma membrane for organization of many signaling proteins (*Brown and London, 1998*). Future studies are needed to directly test whether the plasma membrane/microdomain association of signaling proteins involved in neural differentiation (e.g. insulin and/or bFGF signaling pathways) is disrupted and whether the

phosphorylation of downstream signal transduction factors is reduced in SIRT1 KO mESCs upon induction of neural differentiation. Our findings further suggest that an appropriate content of sphingomyelin, thereby a suitable degree of membrane fluidity, is required to maintain a proper signaling transduction for normal neural differentiation, as too much sphingomyelin resulted from direct sphingomyelin supplementation (*Figure 6C*) or SMPDL3B knockdown (*Figure 8F*) and too little sphingomyelin resulted from SMPDL3B overexpression (*Figure 8B and D*) all impair the expression of neural markers and differentiation.

Our observations that *Smpdl3b* promoter is targeted by both c-Myc and EZH2, consequently with 'co-localization' of antagonistic epigenetic marks H3K4me3 and H3K27me3 at the same locus near its TSS (*Figure 4A* and S5A), suggest that *Smpdl3b* in mESCs might be a bivalent promoter-associated gene (*Bernstein et al., 2006*; *Sanz et al., 2008*; *Vastenhouw and Schier, 2012*). The bivalent chromatin domains in ESCs often mark lineage regulatory genes, and it has been proposed that bivalent domains might repress lineage control genes by H3K27me3 during pluripotency while keeping them poised for activation upon differentiation with H3K4me3 (*Vastenhouw and Schier, 2012*). Interestingly, the bivalent chromatin domain on *Smpdl3b* appears to keep it active in mESCs while poising it to be repressed upon differentiation. Moreover, while long-recognized as a transcription repressor through deacetylation of histones, SIRT1 plays an active role in remodeling this bivalent domain by stabilizing c-Myc while restricting EZH2-induced H3K27me3 in mESCs (*Figure 9E*). Our present study reveals that this SIRT1/c-Myc/EZH2-regulated bivalent domain remodeling enables swift membrane remodeling in response to developmental signals, allowing more efficient and synchronous neural differentiation. Premature disruption of this chromatin remodeling complex in mESCs alters membrane fluidity (*Figure 5A*), which may in turn affect developmental signaling transduction (e.g. insulin, bFGF) and impair neural differentiation. Future studies will be needed to elucidate the general role of this bivalent chromatin switch in regulation of pluripotency vs lineage genes, as well as in process of somatic cell reprogramming and/or transformation.

The transcriptional regulation of sphingolipid metabolism and neural differentiation by the SIRT1-Myc axis in mESCs revealed in our present study is very intriguing, as we have previously reported that the same regulatory axis is key for methionine metabolism and maintenance of pluripotency in mESCs (*Tang et al., 2017*). While our data show that accumulation of sphingomyelin in SIRT1 KO mESCs is independently of methionine metabolism (*Figure 1—figure supplement 1B* and *Supplementary file 1*), additional studies are needed to assess how the SIRT1-Myc regulatory axis coordinates diverse metabolic processes to shape stem cell fates in response to different environmental signals. It is also worth noting that SIRT1 KO mESCs have additional lipid metabolic defects, including depletion of monoacyglycerols, accumulation of plasmalogens, acetylcholine, and monohydroxy fatty acids, and altered phospholipids, regardless of medium methionine concentrations (*Figure 1—figure supplement 1B* and *Supplementary file 1*). It will be of importance to evaluate the contribution of these lipid metabolic defects to the observed hypersensitivity of SIRT1 KO embryos to maternal HFD feeding-induced intrauterine growth retardation (*Figure 9A*) in future studies.

Our study has a few important implications. Firstly, the previously uncharacterized transcriptional regulation of SIRT1 on sphingomyelin degradation directly links cellular levels of sphingomyelin and membrane fluidity with cellular energy status, and provides a possible molecular mechanism for the beneficial impacts of SIRT1 small molecule activators and/or $NAD^+$-boosting dietary supplements on human neurodegenerative diseases. Secondly, given the prevalence of obesity and metabolic syndrome in the reproductive population in modern society, the hypersensitivity of SIRT1 KO embryos to maternal HFD feeding-induced intrauterine growth retardation and neurodevelopmental defects suggests that pharmacological activation of SIRT1 by small molecule activators and/or $NAD^+$-boosting dietary supplements might be able to attenuate maternal obesity-associated neonatal complications and defective childhood neurodevelopment (*Iessa and Bérard, 2015*; *Helle and Priest, 2020*; *Tong and Kalish, 2021*).

In summary, our study uncovers a SIRT1-Myc-mediated transcriptional regulation of sphingomyelin degradation that modulates neural differentiation of ESCs. This finding highlights the importance of SIRT1 and its regulation in mESC differentiation and embryonic development, and may have important implications in potential therapeutic strategies again human neurodegenerative diseases and/or maternal obesity-induced adverse developmental outcomes.

## Materials and methods

### Key Resources Table

Please see the appendix.

### Mammalian cell lines

WT ad SIRT1 KO mESCs generated in R1 mESC line have been reported previously (*Tang et al., 2014*; *McBurney et al., 2003*). They were a gift from Dr. Michael McBurney at Ottawa Hospital Research Institute.

ES-E14TG2a (E14) mESC line was purchased from ATCC. WT and SIRT1 KO E14 mESCs were generated by CRISPR/Cas9 mediated gene editing technology using lentivirus carrying either all-in-one empty vector pCRISPR-CG01 vector or pCRISPR-CG01 containing different sgRNAs targeting mouse *Sirt1* gene (GeneCopoeia). Stable single colonies were picked up and screened with immuno-blotting assay using anti-SIRT1 antibodies. Three independent WT and SIRT1 KO E14 mESCs were used for the experiments to minimize the potential off-target effects of each individual line. mESCs stably transfected with pEF1α-FB-dCas9-puro (Addgene #100547) and pEF1α-BirA-V5-neo (Addgene #100548) vectors (dCas9 mESCs) are described previously (*Liu et al., 2017*). WT and SIRT1 KO dCas9 mESCs were generated by a similar strategy as WT and SIRT1 KO E14 mESCs. Different sgRNA sequences targeting promoter region of Smpdl3b (*Supplementary file 4*) were cloned into plasmid pSLQ1651-sgRNA(F + E)-sgGal4 (Addgene #100549) and then were packed into lentivirus. The WT and SIRT1 KO dCas9 mESCs were infected with those lentiviruses to deliver sgRNA into cells, which then guided the dCas9 to bind on the promotor region of *Smpdl3b* and interfere its expression.

WT and SIRT1 KO mESCs with stable Smpdl3b knockdown were generated by infecting WT and SIRT1 KO mESCs with lentivirus containing vector pLKO.1 or constructs expressing shRNAs against Smpdl3b (B11, B12, and C1) (Sigma). WT and SIRT1 KO mESCs with stable overexpression of SMPDL3B were generated with lentivirus carrying vector pLenti-III-ef1α (Addgene) or constructs expressing the full-length SMPDL3B protein. The SMPDL3B H135A mutant was constructed with QuikChange II XL Site-Directed Mutagenesis Kit (Agilent Technologies, 200522–5) against pLenti-III-ef1α-Smpdl3b by using primers described in *Supplementary file 4*. The expression of SMPDL3B in these cells was analyzed by immunoblotting or immunofluorescence staining.

WT and SIRT1 KO mESCs with stable overexpression of WT or a catalytic inactive H355Y mutant (HY) were generated using lentivirus carrying empty vector pLenti-III-ef1α (Addgene) or constructs expressing the full-length WT or HY SIRT1 proteins.

To knockdown the Ezh2 and C-Myc gene expression, WT and Sirt1 KO mESCs were transfected with siRNA against mouse Ezh2 (ThermoFisher, 4390771-s65775; siNeg: 4390843) and c-Myc (Santa Cruz sc-29227; siNeg: siRNA-A sc-37007), with Lipofectamine RNAiMAX (ThermoFisher Scientific). The knockdown of genes expression was evaluated by Quantitative real-time PCR 48 hr after transfection.

All mouse stem cells were maintained on gelatin-coated plates in the ESGRO Complete Clonal Grade Medium (Millipore), and then cultured in the M10 medium (High-glucose DMEM, 10% ES cell FBS, 2 mM L-glutamine, 1 mM sodium pyruvate, 0.1 mM nonessential amino acids, 10 µM 2-mercaptoethanol, and 500 units/ml leukocyte inhibitory factor) for some experiments.

Mel1 hESCs was a gift from Dr. Andrew Elefanty and Edouard Stanley at the University of Queensland, Australia. WT and SIRT1 KO mel1 hESCs were generated by CRISPR/Cas9 mediated gene editing technology using a plasmid containing Cas9, gRNA (AGAGATGGCTGGAATTGTCC (-strand)) and a GFP indicator. The GFP positive cells were purified by flow cytometry, and were either grown on 10 cm dishes with a serial dilution, or in the 96-well plates at a density of 1 cell/well. Single colonies were picked up and subjected to immunofluorescence assay with anti-SIRT1 antibodies. The cell colonies without SIRT1 staining were sequenced to confirm the mutation. Three independent SIRT1 KO mel1 lines were used for the experiments to minimize the potential off target effects of each individual line.

All cell lines were not authenticated at our end. All original and genetically modified ESCs are routinely checked (at least every 6 months) by the NIEHS Quality Assurance Laboratory for

contamination of mycoplasma and other microbes by prolonged culture followed with qPCR-based assays, and they were all free of mycoplasma in our study.

## Mouse models

Whole body SIRT1 knockout, heterozygote and their age-matched littermate WT mice on the C57BL/6J background have been reported before (*Tang et al., 2014*). They were housed in individualized ventilated cages (Techniplast, Exton, PA) with a combination of autoclaved nesting material (Nestlet, Ancare Corp., Bellmore, NY and Crink-l'Nest, The Andersons, Inc, Maumee, OH) and housed on hardwood bedding (Sani-chips, PJ Murphy, Montville, NJ). Mice were maintained on a 12:12 hr light:dark cycle at $22 \pm 0.5°C$ and relative humidity of 40% to 60%. Mice were provided ad libitum autoclaved rodent diet (NIH31, Harlan Laboratories, Madison, WI) and deionized water treated by reverse osmosis. Mice were negative for mouse hepatitis virus, Sendai virus, pneumonia virus of mice, mouse parvovirus 1 and 2, epizootic diarrhea of infant mice, mouse norovirus, *Mycoplasma pulmonis*, *Helicobacter* spp., and endo- and ectoparasites upon receipt and no pathogens were detected in sentinel mice during this study.

Mice were randomly assigned to experimental groups after they were allowed to acclimate for at least one week prior to experiments.

## Metabolomic analysis

WT and SIRT1 KO mESCs were cultured in the complete M10 medium containing 200 µM methionine or methionine restricted M10 medium containing 6 µM methionine for 6 hr (n = 5 biological replicates). WT and SIRT1 KO mel1 hESCs were cultured in serum-free TeSR-E8 medium containing 116 µM methionine or a methionine restricted medium containing 6 µM methionine for 6 hr on Matrigel (n = 4 biological replicates). Cells were then harvested and profiled by metabolomics analysis as previously described (*Tang et al., 2017*). Specifically, about 100 µl of packed cell pellet per sample were submitted to Metabolon, Inc (Durham, NC, USA), where the relative amounts of small molecular metabolites were determined using four platforms of Ultrahigh-Performance Liquid Chromatography-Tandem Mass Spectroscopy (UPLC-MS/MS) as previously described (*Evans et al., 2014*). All methods utilized a Waters ACQUITY UPLC and a Thermo Scientific Q-Exactive high-resolution/accurate mass spectrometer interfaced with a heated electrospray ionization (HESI-II) source and Orbitrap mass analyzer operated at 35,000 mass resolution. Raw data collected from above four analyses were managed by the Metabolon Laboratory Information Management System (LIMS), extracted, peak-identified and QC processed using Metabolon's hardware and software. The hardware and software foundations for these informatics components were the LAN backbone, and a database server running Oracle 10.2.0.1 Enterprise Edition. The final relative abundance of metabolites in each sample was normalized by the respective total protein concentration.

## Sphingolipid analysis

To confirmed the alteration of sphingolipids in SIRT1 KO mESCs, WT and SIRT1 KO mESCs cultured in serum-free ESGRO medium or serum-containing M10 medium were incubated with 5 µM BODIPY FL $C_5$-Sphingomyelin (N-(4,4-Difluoro-5,7-Dimethyl-4-Bora-3a,4a-Diaza-s-Indacene-3-Pentanoyl) Sphingosyl Phosphocholine) (Invitrogen, D3522) or 5 µM BODIPY FL $C_5$-Ceramide (N-(4,4-Difluoro-5,7-Dimethyl-4-Bora-3a,4a-Diaza-s-Indacene-3-Pentanoyl)Sphingosine) (Invitrogen, D3521) together with 5 µM delipidated BSA (as a delivery carrier) in Hanks' buffered salt solution containing 10 mM HEPES (HBSS/HEPES buffer pH 7.4) at 4°C for 30 min to load SM/ceramide. They were then incubated in medium without BODIPY FL $C_5$-Sphingolipid at 37°C for additional 30 min. The intensity of cellular BODIPY FL $C_5$-Sphingomyelin/Ceramide were analyzed by Zeiss LSM 780 UV confocal microscope and by Flow cytometry analysis (Abs 505 nm and Em 511 nm).

To analyze the degradation of sphingomyelins, WT and SIRT1 KO mESCs were pre-load with BODIPY FL $C_5$-Sphingomyelin at 4°C for 30 min. They were then incubated in medium without BODIPY FL $C_5$-Sphingomyelin at 37°C, and the dynamics of loaded BODIPY FL $C_5$-Sphingomyelins in cells were followed using Zeiss LSM 780 UV confocal microscope for additional 12 hr at 37°C.

The relative contents of endogenous sphingomyelins in WT and SIRT1 KO mESCs were analyzed using a commercially available Sphingomyelin Assay Kit (Abcam ab133118) per manufacturer's instruction.

## Quantitative real-time PCR (qPCR)

Total RNAs were isolated from mESCs or mice tissues using Qiagen RNeasy mini-kit (74104). The nuclear and cytoplasmic RNA were separated and enriched by Fisher BioReagents SurePrep Nuclear or Cytoplasmic RNA Purification Kit (Fisher Scientific, BP280550). The cDNA was synthesized with ABI High-Capacity cDNA Reverse Transcription Kits (4374967) and further analyzed with qPCR using iQ SYBR Green Supermix (Biorad). Three biological replications are performed for each experiment and raw data are normalized to the expression level of Rplp0 mRNA levels. The primers used in RT-PCR are listed in (*Supplementary file 4*).

Immunofluorescence analysis mESCs grown on 0.1% gelatin coated coverslips were washed with PBS and fixed with 4% paraformaldehyde (PFA) in PBS (pH 7.4) solution for 20 min at room temperature. They were then incubated with 1% glycine/PBS for 10 min, and cell membrane was permeabilized with 0.3% Triton X-100 in 1% glycine/PBS for 10 min. Cells were further blocked with 1% BSA and 0.05% Tween 20 in PBS for 30 min, incubated with primary antibodies (Key Resources Table) diluted with the blocking solution for overnight at 4°C, then the secondary antibodies Alexa Fluor 488, 594 and 633 (for flow cytometry sorting) (Invitrogen, A-11008, A-11032, A-21052) at 1:1000 in PBS for 1 hr at room temperature. Cells were counterstained for Nuclei with DRAQ5 Fluorescent Probe Solution (Thermal Fisher, 62251) or directly mounted on glass slides with VECTASHIELD Anti-fade Mounting Media (VECTOR LABORATORY) which contains DAPI. The images of stained cells are acquired by Zeiss LSM 780 UV confocal microscope.

## Northern blotting

The probe for Northern Blot hybridization is generated by using North2South Biotin Random Prime Labeling Kit (ThermoFisher, 17075). A 100 ng DNA product, which was synthesized from PCR reaction by using primer pair '5'- CACCGCTAGCGCCACCatgacgctgctcgggtggctgata-3' and 5'-CACCGCGGCCGCtaacacctccagtacgtgcaggct-3'' and the cDNA synthesized from total RNA isolated from mESCs, was used as a template to yield biotin-labeled single strand DNA probe that covers the full-length Smpdl3b mRNA sequence. Forty μg of total RNA isolated from mESC were separated with agarose electrophoresis with RNA Gel Loading Dye (2X) (ThermoFisher, R0641) and Northern-Max 10X Running Buffer (Ambion, AM8671). The separated total RNA samples were further transblotted to positively charged nylon transfer membrane (Cat. 77016) by using S and S TurboBlotter Rapid Downward Transfer System (DAIGGER Scientific) and SSC buffer (ThermoFisher, AM9763). The RNA samples transferred to membrane were further crosslinked by using Stratalinker UV Crosslinker (Model 1800) immediately upon completion of transblotting. The hybridization was performed by using North2South Chemiluminescent Hybridization and Detection Kit (ThermoFisher,17097) and the signaling of positive hybridization on membrane was detected with Chemiluminescent Nucleic Acid Detection Module (ThermoFisher, 89880). All procedures were performed by strictly following protocols for each kit provided by manufacturers.

## Immunoblotting

Cells were washed once with PBS, and were then lysed and scraped with 2 x SDS loading buffer without bromophenol blue. Samples were boiled for 10 min, and quantified. Equal amount of protein lysates was loaded and resolved on SDS-PAGE gel and transferred onto an PVDF membrane (Millipore). Blots were blocked with 5% BSA for 1 hr, incubated with primary antibodies at 4°C overnight, incubated with secondary antibodies for 2 hr, and detected by Odyssey (LI-Cor inc).

## Chromatin immunoprecipitation (ChIP)

To determine the association of RNA polymerase II (Pol II), c-Myc, EZH2, and histone marks H3K4me3 and H3K27me3 on mouse Smpdl3b locus in WT and SIRT1 KO mESCs, cells were fixed, harvested, and sonicated. The resulting sonicated chromatin was processed for immunoprecipitation with respective antibodies (Key Resources Table) as previously described (*Shimbo et al., 2013*).

## RNA-seq analysis

Total RNA was extracted from WT and SIRT1 KO mESCs cultured in ESGRO medium in triplicates. All RNA-seq libraries were prepared with the TruSeq Stranded/Ribo kit (Illumina, San Diego, CA) and sequenced using the pair-end 76 bp protocol at about 520 million reads per library using the

NovaSeq platform (Illumina) per the manufacturer's protocol. Adaptor sequences were removed by Trim Galore (v0.4.4). Then reads were aligned to mouse genome version GRCm38/mm10 using STAR (v2.5.3a) (*Dobin et al., 2013*) with Gencode vM18 annotation. Gene expression values were quantified using RSEM (v1.2.28) (*Li and Dewey, 2011*) and differences in gene expression between experimental conditions were estimated using R package DEseq2 (*Love et al., 2014*) with input reads count from FeatureCounts (*Liao et al., 2014*) in Subread.

GO Biological Process enrichment analysis was performed on 2541 significantly downregulated genes (q < 0.01) in SIRT1 KO mESCs in g:Profiler (https://biit.cs.ut.ee/gprofiler/gost).

Gene set enrichment analysis (GSEA) (v4.1.0) was implemented against all gene ontology (GO) gene sets in Molecular Signatures database (MsigDB v7.2) with 10000 permutations (min size 15, max size 500, FDR q < 0.25).

## Promoter analysis of *Smpdl3b* gene

Potential transcription factors (TFs) on the promoter of Smpdl3b gene were predicted using 'Match' from geneXplain (genexplain.com), in which the association scores, including 'Core Motif Similarity' and 'Weight Matrix Similarity' were calculated (*Chen et al., 2008*). A higher score implies a higher chance of the gene being the target of this TF.

## Inhibition of c-Myc or EZH2 in mESCs

To test the possible roles of transcriptional factor c-Myc and EZH2 in regulation of *Smpdl3b* expression in mESCs, WT and SIRT1 KO mESCs were treated with c-Myc Inhibitor CAS 403811-55-2–Calbiochem (10058-F4) (Millipore Sigma, 475956) at 10 µM or EZh2 inhibitor Tazemetostat (EPZ-6438) (MCE, HY-13803) at indicated concentrations for 48 hr. WT and SIRT1 KO mESCs were also transfected with siRNA against mouse c-Myc (Santa Cruz, sc-29227) or EZH2 (ThermoFisher, 4390771) to knockdown their expression respectively. Cells were collected for 48 hr after transfection for qPCR analysis.

## Site-direct mutagenesis

To further determine the influence of the acetylation status of c-Myc on expression of *Smpdl3b*, mouse c-Myc protein was first cloned into the pHAGE-EF1α-HA-Puro vector (*Zheng et al., 2016*). The major acetylation site of c-Myc protein, K323, was then mutated to either R to mimic deacetylated c-Myc (K323R) or Q to mimic acetylated c-Myc (K323Q) using QuickChange II Site-Directed Mutagenesis Kit (Agilent Technologies, 200522–5) against pHAGE-EF1α-HA-Puro-c-Myc. pHAGE-EF1α-HA-Puro vector, WT, K323R, and K323Q c-Myc constructs were transfected into WT and SIRT1 KO mESCs using Lipofectamine 3000 Reagent (Invitrogen, L3000001). The overexpression of WT and mutant c-Myc protein in cells were confirmed by immunofluorescence staining of transfected cells with anti-c-Myc antibody (Abcam).

To test the importance of the enzymatic activity of SMPDL3B in regulation of sphingomyelin degradation and neural differentiation, mouse WT SMPDL3B protein was first cloned into the pLenti-III-EF1α vector (*Zhang et al., 2011*). An active site H135 was then mutated to A using QuickChange II Site-Directed Mutagenesis Kit (Agilent Technologies, 200522–5) against pLenti-III-EF1α-Smpdl3b.

The sequences of cloning and mutagenesis primers were listed in *Supplementary file 4*.

## Luciferase assay

To directly analyze the transcriptional regulation of *Smpdl3b* expression by c-Myc/SIRT1, firefly luciferase reporters driven by a 3.1 kb mouse *Smpdl3b* promoter fragment (amplified by 5'-tcttacgcgtgctagcccgggctcgagACTCATCCAAAGGACCCAGGTT-3' and 5'- tttatgtttttggcgtcttCCATGGGGCAGCAGGCACACATG-3') containing either wild type (WT) or a mutant c-Myc binding site (E-box) were cloned into pGL3 basic vector. The E-box mutant was constructed with QuikChange II XL Site-Directed Mutagenesis Kit (Agilent Technologies, 200522–5) using primers 5'-cgcgggttcccaccttgtggccagaagatcttctgggcagaactactcgtttggc-3' and 5'- gccaaacgagtagttctgcccagaagatcttctggccacaaggtgggaacccgcg-3'. The WT or mutant plasmids were then transfected into WT and SIRT1 KO mESCs together with the control pRL-TK plasmid (Renilla Luciferase, Promega). Cells were cultured for 48 hr and the luciferase activity was measured using the Dual-Luciferase Reporter

Assay System (Promega, E1751). The final firefly luciferase activity was normalized to the co-expressed renilla luciferase activity.

## Measurement of membrane fluidity

WT and SIRT1 KO mESCs were seeded on glass cover slips and cultured in ESGRO medium overnight. They were then preincubated for 1 hr with or without 2.5 mM MβCD (Sigma-Aldrich, C4555), and stained with 5 µM di-4-ANEPPDHQ (ThermoFisher, D36802) for 30 min. Coverslips were mounted using ProLongGold (Invitrogen), images were acquired on a Zeiss LSM780 confocal microscope. The fluorescent dye was excited at 488 nm and images from 30 individual colonies in each group were acquired at 560 nm for emission from ordered phase and 620 nm for emission from disordered phase. The images were further analyzed ImageJ according to a method described previously (*Owen et al., 2012*).

## In vitro neural differentiation of mESCs

In vitro neural differentiation of WT and SIRT1 KO mESCs were performed essentially as described (*Ying et al., 2003*; *Abranches et al., 2009*). Specifically, WT and SIRT1 KO mESCs maintained in ESGRO Complete PLUS Clonal Grade Medium (Millipore SF001-500P) were gently dissociated with 0.05% Trypsin and plated onto 0.1% gelatin coated cell culture dish at a density of $1 \times 10^4$ cells / $cm^2$ with RHB-A medium (Clontech TaKaba Cellartis, Y40001). Medium was changed every 2 days and cultured for 4 days. Cells were then dissociated with 0.05% Trypsin again and plated into cell culture dish coated with 1 µg/ml Laminin (Sigma, L2020) at a density of $2 \times 10^4$ cells/cm² in RHB-A medium supplemented with 5 ng/ml murine bFGF (Sigma, SRP4038-50UG). Medium was changed again every 2 days for the next 3 weeks. Cell morphology was monitored during the differentiation. The differentiated neural cells were maintained in RHB-A: Neurobasal (ThermoFisher,10888022): B27 Supplement (ThermoFisher,17504044) (1:1:0.02) medium to for a better survival.

## FACS analysis

To quantify the factions of differentiated cells at different stages during in vitro neural differentiation, differentiated cells were harvested with trypsin and fixed with 4% PFA. After fixation, cells are washed with PBS and immunofluorescence stained with different neural markers, and analyzed by FACS.

## Alkaline phosphatase staining

The alkaline phosphatase staining assay was performed using the Alkaline Phosphatase staining kit II as per manufacturer's instructions (Stemgent, Cambridge, MA; cat. no. 00–0055).

## Maternal HFD feeding

To investigate the effects of maternal HFD feeding on embryonic development of control and SIRT1 KO mice, 6- to 8-week-old SIRT1 heterozygous ($Sirt1^{+/-}$) female mice were fed with either control chow diet (NIH-31 contains 4% fat) or a HFD (D12492 contains 36% fat) for 4–8 weeks. They were then bred with age matched $Sirt1^{+/-}$ ± mice fed with chow diet. Early next morning, females with the mating plug (E0.5) were separated from the males into a new cage and put back on the HFD. Embryos from E14.5 and E18.5 were then collected and analyzed. The total feeding time on the HFD is up to 11 weeks. Embryos were collected from at least four dams (pregnant females) for each time point, this sample size was estimated using Chi square based on 100% penetrance of body weight reduction of SIRT1 KO embryos and a 95% power.

All animal procedures were reviewed and approved by National Institute of Environmental Health Sciences Animal Care and Use Committee, under an Animal Study Proposal number 2017–0008 STL. All animals were housed, cared for, and used in compliance with the *Guide for the Care and Use of Laboratory Animals* and housed and used in an Association for the Assessment and Accreditation of Laboratory Animal Care, International (AAALAC) Program.

## Quantification and statistical analysis

Values are expressed as mean ± standard error of mean (SEM) from at least three independent experiments or biological replicates, unless otherwise indicated in the figure legend. Significant

differences between the means were analyzed by the two-tailed, unpaired, Student's t-test, and differences were considered significant at *$p < 0.05$ using Microsoft Office Excel (Version 16.16.27). No methods were used to determine whether the data met assumptions of the statistical approach (e.g. test for normal distribution).

Bioinformatic analyses of RNA-seq data are detailed in RNA-seq analysis section.

## Acknowledgements

We thank Drs. Serena Dudek, Anton Jetten, and members of the Li laboratory for critical reading of the manuscript. We also thank NIEHS Epigenomics Core Facility for performing RNA-seq experiments, and Scotty Dowdy from Cellular and Molecular Pathology Branch and Comparative Medicine Branch of NIEHS for assistance on animal studies.

## Additional information

### Funding

| Funder | Grant reference number | Author |
|---|---|---|
| National Institute of Environmental Health Sciences | Z01 ES102205 | Xiaoling Li |
| National Natural Science Foundation of China | 31730110 and 31661143031 | Zefeng Wang |
| National Institutes of Health | R01CA230631 and R01DK111430 | Jian Xu |
| China Postdoctoral Science Foundation | 2020M681437 | Xiaojuan Fan |

The funders had no role in study design, data collection and interpretation, or the decision to submit the work for publication.

### Author contributions

Wei Fan, Data curation, Formal analysis, Validation, Investigation, Methodology, Writing - original draft, Writing - review and editing; Shuang Tang, Data curation, Formal analysis, Investigation, Methodology, Writing - review and editing; Xiaojuan Fan, Formal analysis, Visualization, Methodology; Yi Fang, Data curation, Investigation, Methodology, Writing - review and editing; Xiaojiang Xu, Leping Li, Jian-Liang Li, Formal analysis, Writing - review and editing; Jian Xu, Resources, Funding acquisition, Writing - review and editing; Zefeng Wang, Conceptualization, Formal analysis, Supervision, Funding acquisition, Visualization, Methodology, Writing - review and editing; Xiaoling Li, Conceptualization, Formal analysis, Supervision, Funding acquisition, Investigation, Writing - original draft, Writing - review and editing

### Author ORCIDs

Jian Xu http://orcid.org/0000-0003-1988-7337
Jian-Liang Li http://orcid.org/0000-0002-6487-081X
Xiaoling Li https://orcid.org/0000-0001-5920-7784

### Ethics

Animal experimentation: All animal procedures were reviewed and approved by National Institute of Environmental Health Sciences Animal Care and Use Committee. All animals were housed, cared for, and used in compliance with the Guide for the Care and Use of Laboratory Animals and housed and used in an Association for the Assessment and Accreditation of Laboratory Animal Care, International (AAALAC) Program. Animal Study Proposal number 2017-0008 STL.

### Decision letter and Author response

Decision letter https://doi.org/10.7554/eLife.67452.sa1

Author response https://doi.org/10.7554/eLife.67452.sa2

## Additional files

### Supplementary files

- Supplementary file 1. Lipid alterations in WT and SIRT1 KO mESCs analyzed by metabolomics.
- Supplementary file 2. Accumulation of sphingomyelin in both SIRT1 KO hESCs and mESCs.
- Supplementary file 3. Significantly differentially expressed genes between SIRT1 KO vs WT mESCs.
- Supplementary file 4. Oligonucleotides used in the study.
- Transparent reporting form

### Data availability

The RNA-seq (RNA-seq) data has been deposited to Gene Expression Omnibus under the accession number GSE163920 ( https://www.ncbi.nlm.nih.gov/geo/query/acc.cgi?acc=GSE163920). Additional information on DEGs is available from Supplementary File 3. Metabolomics data (lipid alterations) between WT and SIRT1 KO mESCs is available in Supplementary File 1. Sphingolipid profiles between WT and SIRT1 KO mESCs and hESCs are available in Supplementary File 2. All oligos used in the study are available in Supplementary File 4. All antibodies used in the study are available in the Key Resources Table.

The following dataset was generated:

| Author(s) | Year | Dataset title | Dataset URL | Database and Identifier |
|---|---|---|---|---|
| Fan W, Li X | 2021 | The role of SIRT1 in regulation of transcription and splicing in mouse embryonic stem cells | https://www.ncbi.nlm.nih.gov/geo/query/acc.cgi?acc=GSE163920 | NCBI Gene Expression Omnibus, GSE163920 |

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

# Appendix 1

**Appendix 1—key resources table**

| Reagent type (species) or resource | Designation | Source or reference | Identifiers | Additional information |
|---|---|---|---|---|
| Gene (*M. musculus*) | *Sirt1* | MGI | 93759 | |
| Gene (*M. musculus*) | *Smpdl3b* | MGI | 100340 | |
| Gene (*M. musculus*) | *Myc* | MGI | 17869 | |
| Gene (*M. musculus*) | *Ezh2* | MGI | 14056 | |
| Strain, strain background (*M. musculus*) | SIRT1 KO mouse line | *Tang et al., 2014* | | Whole body SIRT1 KO mice (SIRT1 KO) on C57BL/6J background |
| Strain, strain background (*Escherichia coli*) | Stbl3 | ThermoFisher | C737303 | Competent cells |
| Cell line (*M. musculus*) | WT mESCs | A gift from *McBurney et al., 2003* | | WT R1 mouse embryonic stem cell line |
| Cell line (*M. musculus*) | SIRT1 KO mESCs | A gift from *McBurney et al., 2003* | | SIRT1 KO R1 mouse embryonic stem cell line generated by gene targeting technology |
| Cell line (*M. musculus*) | ES-E14TG2a (E14) | ATCC | CRL-1821; RRID: CVCL_9108 | Mouse embryonic stem cell line |
| Cell line (*M. musculus*) | ES-E14TG2a (E14), WT | This paper | | WT E14 mouse embryonic stem cell line, generated with a CRISPR/Cas9 vector (pCRISPR-CG01) (Materials and Methods, Mammalian cell lines) |
| Cell line (*M. musculus*) | ES-E14TG2a (E14), SIRT1 KO | This paper | | SIRT1 KO E14 mouse embryonic stem cell line, generated with pCRISPR-CG01-mSirt1 (Materials and Methods, Mammalian cell lines) |
| Cell line (*M. musculus*) | dCas9 mESCs | *Liu et al., 2017* | | Mouse embryonic stem cell line stably expressing a dox-inducible dCas9 and BirA-V5 |
| Cell line (*M. musculus*) | dCas9 mESCs, WT | This paper | | WT dCas9 mESCs generated with a CRISPR/Cas9 vector (pCRISPR-CG01) (Materials and Methods, Mammalian cell lines) |

*Continued on next page*

*Appendix 1—key resources table continued*

| Reagent type (species) or resource | Designation | Source or reference | Identifiers | Additional information |
|---|---|---|---|---|
| Cell line (*M. musculus*) | dCas9 mESCs, SIRT1 KO | This paper | | SIRT1 KO dCas9 mESCs generated with pCRISPR-CG01-mSirt1 (Materials and Methods, Mammalian cell lines) |
| Cell line (*Homo sapiens*) | mel1 hESCs | Dr. Andrew Elefanty and Edouard Stanley at the University of Queensland, Australia | NIH Registration number: 0139 | Human embryonic cell line |
| Cell line (*Homo sapiens*) | mel1 hESCs, SIRT1 KO | This paper | | SIRT1 KO human embryonic cell line generated with Cas9WT-hSirt1 (Materials and Methods, Mammalian cell lines) |
| Transfected construct (*M. musculus*) | pCRISPR-CG01 | GeneCopoeia | RRID:Addgene_74293 | Backbone vector of all-in-one sgRNA clones for mouse Sirt1 gene |
| Transfected construct (*M. musculus*) | pCRISPR-CG01-mSirt1 | GeneCopoeia | MCP000956-CG01-3-B-a; b | All-in-one sgRNA constructs targeting mouse Sirt1 gene |
| Transfected construct (*H. sapiens*) | Cas9WT-hSirt1 | Horizon Discovery (gift) | | All-in-one sgRNA construct targeting human Sirt1 gene |
| Transfected construct (*M. musculus*) | pSLQ1651-sgRNA (F + E)-sgGal4 | Addgene (a gift from *Liu et al., 2017*) | http://n2t.net/addgene: 100549; RRID:Addgene_100549 | backbone vector of constructs of sgRNAs targeting the promoter region of Smpdl3b gene |
| Transfected construct (*M. musculus*) | pSLQ1651-sgRNA (F + E)-sgGal4-327; −426;+517;+528 | This paper | | Constructs of sgRNAs targeting the promoter region of Smpdl3b gene (Materials and Methods, Mammalian cell lines) |
| Transfected construct (*M. musculus*) | pLKo.1(plasmid) | Sigma | SHC001; RRID: Addgene_10879 | Backbone vector of Smpdl3b gene silencing shRNA plasmid constructs |
| Transfected construct (*M. musculus*) | pLKo.1-B11; B12; C1 (plasmid) | Sigma | TRCN0000099683; TRCN0000311166; TRCN0000304921 | Smpdl3b gene silencing shRNA plasmid constructs |
| Transfected construct (*M. musculus*) | pLenti-III-ef1α (plasmid) | Addgene (a gift from *Zhang et al., 2011*) | http://n2t.net/addgene: 27964; RRID:Addgene_27964 | Backbone vector of Sirt1WT and HY mutant, Smpdl3b and H135A mutant expression constructs |
| Transfected construct (*M. musculus*) | pLenti-III-ef1α-Sirt1 WT (plasmid) | This paper | | Sirt1WT expression construct (Materials and Methods, Mammalian cell lines) |
| Transfected construct (*M. musculus*) | pLenti-III-ef1α-Sirt1 H355Y (plasmid) | This paper | | Sirt1 H335Y mutant expression construct (Materials and Methods, Mammalian cell lines) |
| Transfected construct (*M. musculus*) | pLenti-III-ef1α-Smpdl3b (plasmid) | This paper | | Smpdl3b expression construct (Materials and Methods, Mammalian cell lines) |
| Transfected construct (*M. musculus*) | pLenti-III-ef1α-Smpdl3b H135A Mut (plasmid) | This paper | | Smpdl3b H135A mutant expression construct (Materials and Methods, Mammalian cell lines) |

*Continued on next page*

*Appendix 1—key resources table continued*

| Reagent type (species) or resource | Designation | Source or reference | Identifiers | Additional information |
|---|---|---|---|---|
| Transfected construct (*M. musculus*) | pHAGE-ef1α-HA-Puro | *Zheng et al., 2016* (a gift from Guang Hu) | | backbone vector of c-Myc WT, K323R and K323Q mutant constructs |
| Transfected construct (*M. musculus*) | pHAGE-ef1α-HA-Puro-c-Myc-WT | This paper | | c-Myc expression construct (Materials and Methods, Site-direct mutagenesis) |
| Transfected construct (*M. musculus*) | pHAGE-ef1α-HA-Puro-c-Myc-K323R | This paper | | c-Myc K323R mutant expression construct (Materials and Methods, Site-direct mutagenesis) |
| Transfected construct (*M. musculus*) | pHAGE-ef1α-HA-Puro-c-Myc-K323Q | This paper | | c-Myc K323Q mutant expression construct (Materials and Methods, Site-direct mutagenesis) |
| Recombinant DNA reagent | pGL3 basic (plasmid) | Promega | E1751 | Backbone vector of luciferase assay plasmids constructs |
| Recombinant DNA reagent | pGL3-Smpdl3b (plasmid) | This paper | | Smpdl3b gene promoter region luciferase assay plasmids construct (Materials and Methods, Luciferase assay) |
| Recombinant DNA reagent | pGL3-Smpdl3b E-box Mut (plasmid) | This paper | | Smpdl3b gene promoter region E-box mutation luciferase assay plasmids construct (Materials and Methods, Luciferase assay) |
| Antibody | Anti-SIRT1(Rabbit polyclonal) | Cell Signaling | 2028; RRID:AB_1196631 | IF(1:400) WB(1:1000) |
| Antibody | Anti-SIRT1(Rabbit polyclonal) | Cell Signaling | 2493; RRID:AB_2188359 | IF(1:200) WB(1:1000) |
| Antibody | Anti-SMPDL3B (Mouse monoclonal) | Santa Cruz | sc-137113; RRID:AB_2193525 | IF (1:400) WB(1:1000) |
| Antibody | Anti-SMPDL3B (Rabbit polyclonal) | Thermo Fisher Scientific | PA5-40798; RRID:AB_2606294 | IF(1:400) |
| Antibody | Anti-Nanog (Rabbit polyclonal) | Millipore | ab5731; RRID:AB_2267042 | WB(1:1000) |
| Antibody | Anti-OCT3/4(Goat polyclonal) | Santa Cruz | sc8628; RRID:AB_653551 | WB(1:1000) |
| Antibody | Anti-NESTIN (Mouse monoclonal) | Thermo Fisher Scientific | MA1-110; RRID:AB_2536821 | IF(1:400) WB(1:1000) |
| Antibody | anti-Sox1(Rabbit polyclonal) | Thermo Fisher Scientific | PA5-23370; RRID:AB_2540893 | IF(1:400) |
| Antibody | Anti-Tau (Mouse monoclonal) | Abcam | ab80579; RRID:AB_1603723 | IF(1:400) |
| Antibody | Anti-Tyrosine Hydroxylase (Rabbit polyclonal) | Abcam | ab137721; RRID:AB_2891220 | IF(1:400) |
| Antibody | Anti-beta III Tubulin (Rabbit polyclonal) | Abcam | ab18207; RRID:AB_444319 | IF(1:400) FACS: 1:400 |
| Antibody | Anti-GABA (Mouse monoclonal) | Millipore Sigma | SAB4200721; RRID:AB_2891218 | IF(1:400) FACS: 1:400 |
| Antibody | Anti-CNPase Monoclonal Antibody (Mouse monoclonal) | Thermo Fisher Scientific | MA5-31374; RRID:AB_2787011 | FACS(1:400) |

*Continued on next page*

*Appendix 1—key resources table continued*

| Reagent type (species) or resource | Designation | Source or reference | Identifiers | Additional information |
|---|---|---|---|---|
| Antibody | Anti-Neurofilament heavy polypeptide (Rabbit polyclonal) | Abcam | ab8135; RRID:AB_306298 | IF(1:400) |
| Antibody | Anti-CRISPR/Cas9 (Mouse monoclonal) | Millipore Sigma | SAB4200701; RRID:AB_2891217 | WB (1:1000) |
| Antibody | Anti-Ezh2 (Rabbit monoclonal) | Cell Signaling | 5246; RRID:AB_10694683 | CHIP(1:100) WB(1:1000) |
| Antibody | Anti-C-Myc (Rabbit monoclonal) | Cell Signaling | 5605; RRID:AB_1903938 | CHIP(1:100) |
| Antibody | Anti-C-Myc (Mouse monoclonal) | Abcam | ab32; RRID:AB_303599 | IF(1:400) |
| Antibody | Anti-Pol II (Mouse monoclonal) | Santa Cruz | sc-56767; RRID:AB_785522 | ChIP(1:100) |
| Antibody | Anti-H3K9Me2 (Rabbit polyclonal) | Active Motif | 39753; RRID:AB_2793331 | ChIP (1:100) |
| Antibody | Anti-H3K4Me3(Rabbit polyclonal) | Active Motif | 39159; RRID:AB_2615077 | ChIP (1:100) |
| Antibody | Anti-H3K27Me3 (Rabbit polyclonal) | Active Motif | 39155; RRID:AB_2561020 | ChIP (1:100) |
| Antibody | Anti-SGMS1 (Rabbit polyclonal) | Sigma -Aldrich | SAB2102133; RRID:AB_10604972 | IF(1:400) |
| Antibody | Anti-SGMS2 (Rabbit polyclonal) | Thermo Fisher Scientific | PA5-26744; RRID: AB_2544244 | IF (1:400) |
| Antibody | Anti-SMPD2 (Rabbit polyclonal) | Proteintech | 15290–1-AP; RRID:AB_2891221 | IF (1:400) |
| Antibody | Anti-SMPD4 (Rabbit polyclonal) | Abcam | Ab133935; RRID: AB_2891216 | IF (1:400) |
| Antibody | Anti-Actin (Mouse monoclonal) | Millipore Sigma | MAB1501; RRID: AB_2223041 | WB(1:10,000) |
| Antibody | Anti-GAPDH (Rabbit monoclonal) | Cell Signaling | 2118S; RRID:AB_561053 | WB(1:5000) |
| Antibody | Goat anti-Rabbit IgG (H + L) | Invitrogen | A-11008; RRID: AB_143165 | Secondary antibody **Alexa Fluor 488** **IF and FACS (1:1000)** |
| Antibody | Goat anti-Mouse IgG (H + L) | Invitrogen | A-11032; RRID: AB_2534091 | Secondary antibody **Alexa Fluor 594** **IF (1:1000)** |
| Antibody | Goat anti-Mouse IgG (H + L) | Invitrogen | A-21052; RRID: AB_2535719 | Secondary antibody **Alexa Fluor 633** **FACS (1:1000)** |
| Sequence-based reagent | Please see ***Supplementary file 4*** | | | |
| Peptide, recombinant protein | murine bFGF | Sigma | SRP4038-50UG | Growth factor supplement for In vitro neural differentiation of mESCs |
| Commercial assay or kit | Sphingomyelin Assay Kit | Abcam | Ab133118 | Endogenous SM enzymatic detection |
| Commercial assay or kit | Alkaline Phosphatase staining kit II | Stemgent | 00–0055 | Alkaline phosphatase staining |

*Continued on next page*

*Appendix 1—key resources table continued*

| Reagent type (species) or resource | Designation | Source or reference | Identifiers | Additional information |
|---|---|---|---|---|
| Commercial assay or kit | RNeasy mini-kit | Qiagen | 74104 | RNA isolation |
| Commercial assay or kit | High-Capacity cDNA Reverse Transcription Kits | ABI | 4374967 | cDNA synthesizes |
| Commercial assay or kit | iQ SYBR Green Supermix | Biorad | 1708887 | Quantitative real-time PCR detection systems |
| Commercial assay or kit | North2SouthTM Biotin Random Prime Labeling Kit | Thermo | 17075 | Northern blot hybridization probes generation |
| Commercial assay or kit | S and S TurboBlotter Rapid Downward Transfer System | DAIGGER SCIENTIFIC | | Northern blot RNA transblotting |
| Commercial assay or kit | North2South Chemiluminescent Hybridization and Detection Kit | Thermo | 17097 | Northern blotting hybridization |
| Commercial assay or kit | Chemiluminescent Nucleic Acid Detection Module | Thermo | 89880 | Northern blotting detection |
| Commercial assay or kit | TruSeq Stranded/Ribo kit | Illumina | | RNA-seq libraries preparation kits |
| Commercial assay or kit | Fisher BioReagents SurePrep Nuclear or Cytoplasmic RNA Purification Kit | Fisher Scientific | BP280550 | Nuclear and cytoplasmic RNA separation |
| Commercial assay or kit | Ezh2 siRNA | Thermofisher | 4390771-s65775 siNeg:4390843 | Ezh2 knockdown |
| Commercial assay or kit | C-Myc siRNA | Santa Cruz | sc-29227; siNeg: siRNA-A sc-37007 | C-Myc knockdown |
| Commercial assay or kit | Lipofectamine RNAiMAX | ThermoFisher | 13778075 | RNAi Transfection Reagent |
| Commercial assay or kit | Lipofectamine 3000 | Invitrogen | L3000001 | Transfection Reagent |
| Commercial assay or kit | QuickChange II Site-Directed Mutagenesis Kit | Agilent Technologies | 200522–5 | Site-specific mutagenesis |
| Commercial assay or kit | Dual-Luciferase Reporter Assay System | Promega | E1960 | Luciferase activity detection |
| Commercial assay or kit | BODIPY FL C5-Sphingomyelin | Invitrogen | D3522 | Dye for Sphingolipid analysis |
| Commercial assay or kit | BODIPY FL C5-Ceramide | Invitrogen | D3521 | Dye for Sphingolipid analysis |
| Chemical compound, drug | di-4-ANEPPDHQ | ThermoFisher | D36802 | Measurement of membrane fluidity |
| Chemical compound, drug | Sphingomyelin | Sigma-Aldrich | S0756 | Sphingomyelin |
| Chemical compound, drug | MβCD | Sigma-Aldrich | C4555 | Methyl-β-cyclodextrin |

*Appendix 1—key resources table continued*

| Reagent type (species) or resource | Designation | Source or reference | Identifiers | Additional information |
|---|---|---|---|---|
| Chemical compound, drug | Tazemetostat; EPZ-6438 | MCE | HY-13803 | EZh2 inhibitor |
| Chemical compound, drug | CAS 403811-55-2–Calbiochem; 10058-F4 | Millipore Sigma | 475956 | c-Myc Inhibitor |
| Software, algorithm | Trim Galore (v0.4.4) | *Krueger, 2021* | RRID:SCR_011847 | RNA-seq analysis |
| Software, algorithm | STAR aligner (v2.5.3a) | *Dobin et al., 2013* | RRID:SCR_004463 | RNA-seq analysis |
| Software, algorithm | RSEM (v1.2.28) | *Li and Dewey, 2011* | RRID:SCR_013027 | RNA-seq analysis |
| Software, algorithm | DEseq2 | *Love et al., 2014* | RRID:SCR_015687 | RNA-seq analysis |
| Software, algorithm | FeatureCounts (version 1.4.6) | *Liao et al., 2014* | RRID:SCR_012919 | RNA-seq analysis |
| Software, algorithm | g:Profiler | https://biit.cs.ut.ee/gprofiler/gost | RRID:SCR_006809 | RNA-seq analysis, Pathway enrichment |
| Software, algorithm | Gene set enrichment analysis (GSEA) (v4.1.0) | https://www.gsea-msigdb.org | | RNA-seq analysis, Pathway enrichment |
| Software, algorithm | Match | geneXplain (genexplain.com) | | Promoter analysis of Smpdl3b gene |
| Software, algorithm | Metabolon Laboratory Information Management System (LIMS) | Metabolon, Inc | | Metabolomic analysis |
| Software, algorithm | Microsoft Office Excel (version 16.16.27) | Microsoft | RRID:SCR_016137 | Data graphing and statistical analysis |
| Software, algorithm | Prism 9 (v9.0.0) | Graphpad | RRID:SCR_000306 | Data graphing and statistical analysis |
| Others | VECTASHIELD Antifade Mounting Media | Vector Laboratory | H-1800 | Contains DAPI |
| Others | DRAQ5 Fluorescent Probe Solution | Thermal Fisher | 62251 | Cells nuclei counterstaining |

