## [Decision Letter]

**Acceptance summary:**

This paper will be of broad interest to cell biologists and advances the current understanding of the relationship between lipid metabolism and stem cell function. The paper defines mechanistic connections between SIRT1, sphingolipids, and stem cell differentiation. The data generated from multiple complementary experimental approaches are of high quality and convincingly support the claims made.

**Decision letter after peer review:**

Thank you for submitting your article "SIRT1 regulates sphingolipid metabolism and neural differentiation of mouse embryonic stem cells through c-Myc- SMPDL3B" for consideration by *eLife*. Your article has been reviewed by 2 peer reviewers, one of whom is a member of our Board of Reviewing Editors, and the evaluation has been overseen by Matt Kaeberlein as the Senior Editor. The reviewers have opted to remain anonymous.

The reviewers appreciated the interest of the findings and the high quality the experiments. Most of the reviewers' specific comments can be addressed with text clarifications and discussions. Major new experimental studies are not required.

Essential revisions:

1. A discussion about how impairing the degradation of sphingomyelin might influence the differentiation of ESCs (signaling, protein localization in membranes etc.) would be a valuable addition to the manuscript.

2. With respect to Figure 6C please comment on why SM treatment increases protein levels of Nestin in WT ESC but decreases protein levels of Nestin in Sirt1 KO ESCs. Would removing SM with SMase also lower the protein levels of Nestin in Sirt1 ESCs?

3. The labeling in Figure 8 is very confusing and there are apparent inconsistencies between the data presented here and the data in Figure 6. The four panels in the top left of Figure 8B (staining for TUBB3) seem to show more TUBB3 (or at least equal amounts) in the Sirt1 KO cells, and reintroduction of Smpdl3 to the Sirt1 KO cells appears to lower TUBB3. This is inconsistent with the text. TUBB3 staining is also presented in the four panels on the top right of Figure 8B. The label under the graph says 6 weeks induction but there is no indication in the text or the methods what this means. The TUBB3 staining in these four panels (top right of Figure 8B) shows that TUBB3 is lower in the Sirt1 KOs (empty vector). This set of panels is consistent with the text but its not clear what the difference is between the TUBB3-stained panels on the top right and the TUBB3-stained panels on the top left. In the bottom left four panels of Figure 8B, Nestin staining is lower in the Sirt1 KOs compared to WT treated with vector control. This is inconsistent with Figure 6C which shows that Nestin is higher in the Sirt1 KOs.

4. The authors present strong evidence that sphingomyelin degradation is mediated via SMPDL3B, but its enzymatic activity has not been directly investigated. Could SMPDL3B phosphodiesterase activity be detected in cellular assays or the requirement for its enzymatic activity demonstrated in rescue experiments?

5. Was Smpdl3b expression kinetics determined in the neural differentiation experiment shown in Figure 7B? The expression pattern from pluripotent to more committed cells (which I expect to follow SIRT1) would be informative for potential temporal effects during neural development.

6. The authors mainly focus on the murine system, but also provide evidence for accumulation of sphingomyelin in human ES cells upon SIRT1 knockout. However, this effect seems to be less striking in human than in murine cells (Table S2). Is there a reason why the metabolomics experiment was performed in presence of serum for the murine and in serum-free conditions for the human datasets? Could this be a reason for the observed discrepancy?

7. In Figure S5, the functional annotation analysis has been performed using all 2839 genes downregulated in SIRT1 KO ES cells, without considering their significance (padj). I suggest performing this analysis considering only genes that show a significant reduction in SIRT1 KO. It could also make sense to intersect significantly down-regulated genes with genes involved in lipid metabolism to identify potential candidates contributing to SIRT1/SMPDL3B-function.

---

## [Author Response]

Essential revisions:1. A discussion about how impairing the degradation of sphingomyelin might influence the differentiation of ESCs (signaling, protein localization in membranes etc.) would be a valuable addition to the manuscript.

Many thanks for this suggestion. We added a paragraph of discussion on the possible molecular mechanisms underlying defective ESC differentiation caused by impaired sphingomyelin degradation (Page 17, Line 403-415):

“How impaired degradation of sphingomyelin might influence the differentiation of ESCs remains unclear. […] Future studies are needed to directly test whether the plasma membrane/microdomain association of signaling proteins involved in neural differentiation (e.g. insulin and/or bFGF signaling pathways) is disrupted and whether the phosphorylation of downstream signal transduction factors is reduced in SIRT1 KO mESCs upon induction of neural differentiation.”.

2. With respect to Figure 6C please comment on why SM treatment increases protein levels of Nestin in WT ESC but decreases protein levels of Nestin in Sirt1 KO ESCs. Would removing SM with SMase also lower the protein levels of Nestin in Sirt1 ESCs?

Thanks for the insightful question. Based on our observations in this study, we believe that an appropriate content of sphingomyelin, thereby a suitable degree of membrane fluidity, is required to maintain a proper signaling transduction for normal maintenance of pluripotent stem cells as well as for normal neural differentiation. Specifically, for the results in Figure 6C, since WT ESCs have a low level of sphingomyelin, we speculate that supplementation with exogenous sphingomyelin increases membrane contents of sphingomyelin, which may in turn increase the microdomain formation and inappropriately enhance neural differentiation signaling. SIRT1 KO mESCs, on the other hand, already have accumulation of sphingomyelin. Further treatment with exogenous sphingomyelin may disrupt the membrane structure, leading to impaired expression of Nestin. Yes, removing sphingomyelin by SMPDL3B in SIRT1 KO mESCs could lower the protein levels of Nestin. As shown in our new Figure 6D, overexpression of WT but not H135A mutant SMPDL3B reduced the expression of Nestin in SIRT1 KO mESCs. We added a few sentences in the Discussion (Page 18, Line 415-421) on the importance of an appropriate sphingomyelin content/membrane fluidity in pluripotent stem cells and neural differentiation:

“Our findings further suggest that an appropriate content of sphingomyelin, thereby a suitable degree of membrane fluidity, is required to maintain a proper signaling transduction for normal neural differentiation, as too much sphingomyelin resulted from direct sphingomyelin supplementation (Figure 6C) or SMPDL3B knockdown (Figure 8F) and too little sphingomyelin resulted from SMPDL3B overexpression (Figure 8B and 8D) all impair the expression of neural markers and differentiation.”

3. The labeling in Figure 8 is very confusing and there are apparent inconsistencies between the data presented here and the data in Figure 6. The four panels in the top left of Figure 8B (staining for TUBB3) seem to show more TUBB3 (or at least equal amounts) in the Sirt1 KO cells, and reintroduction of Smpdl3 to the Sirt1 KO cells appears to lower TUBB3. This is inconsistent with the text. TUBB3 staining is also presented in the four panels on the top right of Figure 8B. The label under the graph says 6 weeks induction but there is no indication in the text or the methods what this means. The TUBB3 staining in these four panels (top right of Figure 8B) shows that TUBB3 is lower in the Sirt1 KOs (empty vector). This set of panels is consistent with the text but its not clear what the difference is between the TUBB3-stained panels on the top right and the TUBB3-stained panels on the top left. In the bottom left four panels of Figure 8B, Nestin staining is lower in the Sirt1 KOs compared to WT treated with vector control. This is inconsistent with Figure 6C which shows that Nestin is higher in the Sirt1 KOs.

We apologize for the confusion caused by data in Figure 6 and Figure 8.

Firstly, please let us to clarify that the data Figure 6 is from undifferentiated mESCs, in which deletion of SIRT1 leads to increased expression of neural differentiation markers (which should be silenced at the pluripotent stage). On the other hand, data in Figure 8 is from cells after 4 weeks of neural differentiation (excepted for the right four panels in Figure 8B). Our data indicate that although SIRT1 KO mESCs have an increased expression of neural marker at the stem cell stage, upon differentiation, the induction of these markers is blunted. Therefore, both the reduction of pluripotency markers and the induction of NSC/neural differentiation factors were significantly blunted when SIRT1 was deleted, which is consistent with our qPCR data in Figure 7B.

Secondly, the two TUBB3 staining in Figure 8B are from different time points after differentiation. The left four panels are after regular 4 weeks of neural differentiation, while the right four panels are after 6 weeks of neural differentiation for a better morphological analysis. The primary purpose of the TUBB3 staining was to show the atypical morphology of these TUBB3 positive neurons differentiated from SIRT1 KO mESCs. They were not meant to show their overall expression levels. The overall expression of TUBB3 in 4-week neural differentiation culture was quantified by FACS in original Figure 8D (new Figure 8C).

To avoid further confusion, we rearranged Figure 8 in the revision. We split the original Figure 8B into two parts, with TUBB3 staining as new Figure 8B for neuronal morphological alteration. The Nestin and GABA staining images are now new Figure 8D for neural marker expression alteration.

4. The authors present strong evidence that sphingomyelin degradation is mediated via SMPDL3B, but its enzymatic activity has not been directly investigated. Could SMPDL3B phosphodiesterase activity be detected in cellular assays or the requirement for its enzymatic activity demonstrated in rescue experiments?

Great suggestion. To address this question, we generated a catalytic inactive mutant of SMPDL3B (H135A), then overexpressed it into WT and SIRT1KO mESCs and analyzed its impact on cellular levels of sphingomyelin and expression of Nestin in mESCs. Our data indicate that the activity of SMPDL3B is required to reduce cellular levels of sphingomyelin in WT, particularly SIRT1 KO, mESCs (Figure 3D, 3E, and 3F). Its activity is also required to rescue (repress) the abnormal activation of Nestin in undifferentiated SIRT1 KO mESCs (Figure 6D). Unfortunately, due to the visa renewal issue of Dr. Fan, we did not have sufficient time to test whether the enzymatic activity of SMPDL3B is required to rescue the neural differentiation defects in SIRT1 KO mESCs.

5. Was Smpdl3b expression kinetics determined in the neural differentiation experiment shown in Figure 7B? The expression pattern from pluripotent to more committed cells (which I expect to follow SIRT1) would be informative for potential temporal effects during neural development.

Thanks for the suggestion. We analyzed the mRNA levels of *Smpdl3b* during the 4-week of neural differentiation, and the result is shown in new Figure 7B. Yes, the expression pattern of *Smpdl3b* is similar to that of SIRT1 during in vitro neural differentiation.

6. The authors mainly focus on the murine system, but also provide evidence for accumulation of sphingomyelin in human ES cells upon SIRT1 knockout. However, this effect seems to be less striking in human than in murine cells (Table S2). Is there a reason why the metabolomics experiment was performed in presence of serum for the murine and in serum-free conditions for the human datasets? Could this be a reason for the observed discrepancy?

Yes, the reviewer is correct that the accumulation of sphingomyelin in SIRT1 KO mESCs is much more striking than that in SIRT1 KO hESCs. We also believe this difference is primarily due to the presence of serum in mESC culture, as serum is the major source of lipids for cultured cells.

The reason for why mESCs were cultured in serum-containing medium while hESCs were cultured in serum-free medium for the metabolomics experiments was primarily technical. As mentioned in Figure 1, the metabolomics experiments were first performed to assess the impact of methionine on the maintenance of SIRT1 KO ESCs for one of our previous studies (30). We had to use serum-containing medium for methionine restriction in mESCs due to the unavailability of methionine-restricted serum-free ESGRO medium. hESCs, on the other hand, can only be maintained in serum-free medium as serum will induce their differentiation. We were able to make methionine-restricted serum-free medium for hESCs.

7. In Figure S5, the functional annotation analysis has been performed using all 2839 genes downregulated in SIRT1 KO ES cells, without considering their significance (padj). I suggest performing this analysis considering only genes that show a significant reduction in SIRT1 KO. It could also make sense to intersect significantly down-regulated genes with genes involved in lipid metabolism to identify potential candidates contributing to SIRT1/SMPDL3B-function.

Thanks again for the suggestion. We reanalyzed our RNA-seq data using an improved reads count algorithm (FeatureCounts in Subread) and updated the Table S3 with a new DEG list generated with q (padi)<0.01 as a cutoff. GO analysis of the 2541 significantly downregulated genes in SIRT1 KO mESCs revealed that pathways involved in cell surface receptor signaling pathway and intracelluar signal transduction were among the most significantly disrupted GO Biological Process (GO BP) in SIRT1 KO mESCs when compared with WT mESCs (new Figure S6A). Interestingly, *Smpdl3b* was grouped in three overlapping pathways involved in phosphorus metabolism (highlighted in red). Future studies are needed to explore the intriguing link between sphingomyelin degradation and phosphorus metabolism, including phosphate transport, production, and protein phosphorylation-mediated signaling.